# Structures and Biological Activities of Secondary Metabolites from *Xylaria* spp.

**DOI:** 10.3390/jof10030190

**Published:** 2024-02-29

**Authors:** Weikang Chen, Miao Yu, Shiji Chen, Tianmi Gong, Linlin Xie, Jinqin Liu, Chang Bian, Guolei Huang, Caijuan Zheng

**Affiliations:** 1Key Laboratory of Tropical Medicinal Resource Chemistry of Ministry of Education, College of Chemistry and Chemical Engineering, Hainan Normal University, Haikou 571158, China; 18971192012@163.com (W.C.); yumiaonpc@126.com (M.Y.); chenshijinpc@126.com (S.C.); 19971376959@163.com (T.G.); xll17716396815@163.com (L.X.); 17800717095@163.com (J.L.); 15522857810@163.com (C.B.); 2Key Laboratory of Tropical Medicinal Plant Chemistry of Hainan Province, Haikou 571158, China

**Keywords:** *Xylaria* sp., secondary metabolites, bioactivity

## Abstract

The fungus genus *Xylaria* is an important source of drug discoveries in scientific fields and in the pharmaceutical industry due to its potential to produce a variety of structured novel and bioactive secondary metabolites. This review prioritizes the structures of the secondary metabolites of *Xylaria* spp. from 1994 to January 2024 and their relevant biological activities. A total of 445 new compounds, including terpenoids, nitrogen-containing compounds, polyketides, lactones, and other classes, are presented in this review. Remarkably, among these compounds, 177 compounds show various biological activities, including cytotoxic, antimicrobial, anti-inflammatory, antifungal, immunosuppressive, and enzyme-inhibitory activities. This paper will guide further investigations into the structures of novel and potent active natural products derived from *Xylaria* and their potential contributions to the future development of new natural drug products in the agricultural and medicinal fields.

## 1. Introduction

The fungus genus *Xylaria*, belonging to the family Xylariaceae, is a fungus widely distributed in both marine and terrestrial environments. Most of the genus *Xylaria* is saprophytic, digesting rotten wood, bark, feces, and other organic matter; similar to most saprophytic fungi, it can produce a variety of species. *Xylaria* species are famous for producing structured novel and potent bioactive secondary metabolites. The secondary metabolites obtained from the fungus genus *Xylaria* have high biological activity, including antibacterial, antioxidant, and cytotoxic activities [1,2,3,4]. The fungus genus *Xylaria* also has the potential to be used as a bioremediation agent and enzymatic degradation agent in industrial and agricultural fields [5,6,7].

The *Xylaria* fungi are producers of structurally diverse and biologically active compounds. As of 2020, 245 bioactive compounds (118 new compounds), including sesquiterpenoids, terpenoids, cytochalasins, alkaloids, polyketides, and aromatic compounds, have been isolated from the genus *Xylaria*. These compounds displayed a wide range of biological activities, comprising antibacterial, antifungal, anticancer, antimalarial, anti-inflammatory, and *α*-glucosidase inhibitory activities. Many of these compounds exhibit a strong potential to be expanded into novel drugs [8,9]. The secondary metabolites with novel structures and diverse bioactivities from *Xylaria* have continued to attract great attention from chemists, agricultural chemists, and pharmacologists.

The current review summarizes the chemical diversity and bioactivities of 445 new compounds isolated from *Xylaria* species from 1994 to January 2024. Structurally, they are classified into terpenoids (133 compounds), nitrogen-containing compounds (112 compounds), polyketides (70 compounds), lactones (76 compounds), and other compounds (54 compounds). Among them, 177 compounds display a wide range of biological activities, including cytotoxic, antimicrobial, anti-inflammatory, antifungal, antiplasmodial, immunosuppressive, and enzyme-inhibitory activities. This review summarizes the sources, chemical structures, and biological activities of 445 new compounds reported in the genus *Xylaria* in the past 30 years (between 1994 and January 2024), in order to provide a reasonable and reliable theoretical basis for the future development of new natural drug products in the agricultural and medicinal fields.

## 2. Structural and Biological Activity Studies

### 2.1. Terpenoids

Terpenoids usually comprise isoprene or isopentane unit structures. A total of 133 new terpenoids were discovered from the genus of *Xylaria* sp., including 84 sesquiterpenes, 43 diterpenes, and six triterpenoids. Remarkably, 38 of them showed cytotoxic activities, antibacterial activities, antifungal activities, *α*-glucosidase inhibitory activities, and so on.

#### 2.1.1. Sesquiterpenes

One new sesquiterpene, 13,13-dimethoxyintegric acid (**1**), was isolated from a dead branch-derived fungus *Xylaria* sp. V-27 (Figure 1). Compound **1** promoted growth-restoring activity against a mutant yeast strain (*Saccharomyces cerevisiae* zds1Δ erg3Δ pdr1Δ pdr3Δ) and inhibited the degranulation of rat basophilic leukemia RBL-2H3 cells stimulated by Immunoglobulin E+2,4-dinitrophenylated-bovine serum albumin (IgE+DNP-BSA), thapsigargin, and A23187, with half maximal inhibitory concentration (IC_50_) values of 42.2, 21.2, and 37.5 μM, respectively [10]. Three new compounds, including 10-hydroxythujopsene (**2**), akotriol (**3**), and xylaritriol (**4**), were isolated from the *Litsea akoensis*-derived fungus [11]. Six new sesquiterpenes, including nigriterpenes A–F (**5**–**10**) with eremophilane skeletons, were obtained from the termite nest-derived *Xylaria nigripe*. Among them, nigriterpene C (**7**) showed concentration-dependent inhibition of lipopolysaccharide-induced inducible nitric oxide synthase (iNOS), cyclooxygenase-2 (COX-2) expression, and nitric oxide (NO) production in murine brain microglial BV-2 cells, with IC_50_ values of 8.1, 16.6, and 21.7 μM, respectively [12]. Two new sesquiterpenoids, including polymorphines A and B (**11** and **12**) with drimane skeletons, were separated from the fungus *Xylaria polymorpha* (Pers.: Fr.) Grer. Compound **12** showed acetylcholinesterase (AChE) inhibitory activity (inhibition rate of 34.3%; final reaction concentration of 50 µg/mL) and also showed weak *α*-glucosidase inhibitory activity, with an IC_50_ value of 543.8 µM [13]. Three new compounds, including xylaric acids A–C (**13**–**15**), were isolated from the termite nest-derived fungus *Xylaria* sp. [14]. Two new eremophilane sesquiterpenes, including eremoxylarins A (**16**) and B (**17**), were obtained from *Xylaria* sp. (YUA-026). The fungus YUA-026 was collected from twigs and petioles of Mt. Takadate, Japan. Compounds **16** and **17** displayed activity against *S. aureus*, with minimum inhibitory concentration (MIC) values of 12.5 and 25 μg/mL, respectively, and against *Pseudomonas aeruginosa* with MIC values of 6.25 and 12.5 μg/mL, respectively [15]. A new compound, eremoxylarin C (**18**), was isolated from the wood decay fungus *Xylaria allantoidea* BCC 23163. Also, **18** showed inhibitory activity against *Plasmodium falciparum* K1 and human small-cell lung cancer (NCI-H187) cells, with IC_50_ values of 3.1 and 6.7 μg/mL, respectively [16]. Seven new compounds, including eremoxylarins D–J (**19**–**25**) with eremophilane skeletons, were separated from the coculture fermentation of *Xylaria hypoxylon* and *Dendrothyrium variisporum*. The fungus *X. hypoxylon* was derived from the lichen *Rhizocarpon geographicum*. Compounds **19**, **21**, **22**, and **24** exhibited activity against three Gram-positive bacteria including *Staphylococcus aureus*, methicillin-resistant *S. aureus* (MRSA), and *S. epidermidis*, with MIC values from 0.39 to 12.50 μg/mL. Compound **24** was also active against human coronavirus 229E (HCoV-229E) at a concentration nontoxic to human hepatocellular carcinoma (Huh-7) cells (IC_50_, 18.1 μM; the median cytotoxic concentration CC_50_, 46.6 μM) [17]. Ten new compounds, including 10*α*-hydroxyeremophil-7(11)-en-2,3:12,8-diolide (**26**), 1*β*-acetoxy-10*α*,13-dihydroxyeremophil-7(11)-en-12,8*β*-olide (**27**), 1*α*,10*α*-epoxy-2*α*,13-dihydroxyeremophil-7(11)-en-12,8*β*-olide (**28**), 1*α*,10*α*-epoxy-2*β*,13-dihydroxyeremophil-7(11)-en-12,8*β*-olide (**29**), 1*α*,10*α*-epoxy-3*α*,13-dihydroxyeremophil-7(11)-en-12,8*β*-olide (**30**), 1*α*,10*α*-epoxy-3*β*,13-dihydroxyeremophil-7(11)-en-12,8*β*-olide (**31**), 1*α*,10*α*:2*α*,3*α*-diepoxyeremophil-7(11)-en-12,8*β*-olide (**32**), 2-oxo-13-hydroxyeremophila-1(10), 7(11)-dien-12,8*β*-olide(13-hydroxyxylareremophil(**33**), 7-epi-tessaric acid (**34**), and 2*β*-hydroxyeremophila-1(10), 11(13)-dien-12-oic acid (**35**), were isolated from the mangrove-derived fungus *Xylaria* sp. BCC 60405. Compound **31** showed cytotoxic activity against Vero cells (IC_50_, 49.6 μg/mL) [18]. Five new compounds, including xylcarpins A–E (**36**–**40**), were obtained from *Xylaria carpophila* (Pers.) [19]. Four new compounds, including xylarioxides A–D (**41**–**44**), were isolated from the *Azadirachta indica*-derived fungus *Xylaria* sp. YM 311647. Compound **41** exhibited inhibitory activity against two pathogenic fungi including *Curvularia lunata* and *Botrytis cinerea*, with MIC values of 8 and 16 μg/mL, respectively. Compounds **42** and **43** displayed inhibitory activity against two pathogenic fungi including *C. lunata* and *Alternaria alternata*, with the same MIC value of 16 μg/mL [20]. One new eremophilane sesquiterpene, xylareremophil (**45**), was obtained from the leaves of the *Sophora tonkinensis*-derived *Xylaria* sp. (GDG-102). Compound **45** displayed weak antibacterial activity against *Micrococcus luteus* and *Proteus vulgaris*, with the same MIC value of 25 μg/mL [21]. Three new esquiterpenes, including xylarenones A (**46**) and B (**47**) and xylarenic acid (**48**), were obtained from the *Torreya jackii*-derived fungus *Xylaria* sp. (NCY2). Compounds **46**, **47**, and **48** displayed cytotoxicity against HepG2 cell lines, with IC_50_ values of 8.7, 23.8, and 2.63 μg/mL, respectively. Also, **46**, **47**, and **48** showed inhibitory activity against HeLa cells, with IC_50_ values of 27.8, 21.1, and 19.9 μg/mL, respectively [22]. Five new guaiane sesquiterpenes, including (1*S*,2*S*,4*S*,5*S*,7*R*,10*R*)-guaiane-2,10,11,12-tetraol (**49**), (1*S*,2*S*,4*S*,5*S*,7*R*,10*R*)-guaiane-2,4,10,11,12-pentaol (**50**), (1*S*,4*R*,5*S*,7*R*,10*R*)-guaiane-4,5,10,11,12-pentaol (**51**), (1*R*,4*S*,5*R*,7*R*,10*R*)-guaiane-1,5,10,11,12-pentaol (**52**), and (1*R*,4*R*,5*R*,7*R*,10*R*)-11-Methoxyguaiane-4,10,12-triol (**53**), were isolated from the plant *Azadirachta indica*-derived fungus *Xylaria* sp. (YM311647). Compounds **49**–**53** displayed moderate or weak activities against two pathogenic fungi including *Pyricularia oryzae* and *Hormodendrum compactum*, with MIC values ranging from 32 to 256 μg/mL. Compound **52** showed potent antifungal activity against *P. oryzae* with an MIC value of 32 μg/mL. Compounds **51** and **52** exhibited antifungal activity against *H. compactum* with an MIC value of 32 μg/mL. Compounds **51** and **52** showed antifungal activity against *C. albicans* with an MIC value of 32 μg/mL. Also, **51** displayed inhibition activity against *C. albicans*, *A. niger*, and *H. compactum*, with the same MIC value of 64 μg/mL [23]. Eight new eremophilane-type sesquiterpenoids, including 1*β*,7*α*,10*α*-trihydroxyeremophil-11(13)-en-12,8*β*-olide (**54**), 7*α*,10*α*-Dihydroxy-1*β*-methoxyeremophil-11(13)-en-12,8*β*-olide) (**55**), and 1α,10*α*-epoxy-7α-hydroxyeremophil-11(13)-en-12,8*β*-olide (**56**), 1*β*,10α,13-trihydroxyeremophil-7(11)-en-12,8-olide (**57**), 10*β*,13-dihydroxy-1 -methoxyeremophil-7(11)-en-12,8*β*-olide (**58**), mairetolide F (**59**), 1*β*,10*α*-epoxy-13-hydroxyeremophil-7(11)-en-12,8*β*-olide (**60**), and 1*β*,10*α*-epoxy-3-hydroxyeremophil-7(11)-en-12,8*β*-olide (**61**), were purified from the palm *Licuala spinose*-derived fungus *Xylaria* sp. (BCC 21097). Compounds **54**–**56**, with *α*-methylene-*γ*-lactone skeletons, exhibited potent cytotoxicity against human oral epidermal carcinoma KB, human breast cancer MCF-7, NCI-H187, and African green monkey kidney fibroblast Vero cell lines, with IC_50_ values ranging from 0.066 to 15 μM. Compounds **55** and **56** also exhibited antimalarial activity against *P. falciparum* K1 with IC_50_ values of 8.1 and 13 μM, respectively. Also, **56** showed antifungal activity against *C. albicans* with an IC_50_ value of 7.8 μM, suggesting that epoxide functionality may play an important role in antifungal activity [24]. Four new 12,8-eudesmanolides, including 3*α*,4*α*,7*β*-trihydroxy-11(13)-eudesmen-12,8-olide (**62**), 4*α*,7*β*-dihydroxy-3*α*-methoxy-11(13)-eudesmen-12,8-olide (**63**), 7*β*-Hydroxy-3,11(13)-eudesmadien-12,8-olide (**64**), and 13-Hydroxy- 3,7(11)-eudesmadien-12,8-olide (**65**), were isolated from an unidentified seed-derived fungus *Xylaria ianthinovelutina* (Mont.). Compounds **62**–**65** showed cytotoxic activity against NCI-H187, KB, and MCF-7 cell lines, with IC_50_ values varying range from 0.78 to 19.15 µg/mL. Compound **64** also showed antimalarial activity against the *P. falciparumm* K-1 strain with an IC_50_ value of 2.27 μg/mL [25]. Two new presilphiperfolane sesquiterpenes, including 9,15-dihydroxy-presilphiperfolan-4-oic acid (**66**) and 15-acetoxy-9-hydroxy-presilphiperfolan-4-oic acid (**67**), were isolated from the leaves of the *Piper aduncum*-derived fungus *Xylaria* sp. [26]. Three new eremophilane sesquiterpenes (**68**–**70**) were isolated from the mangrove-derived fungus *Xylaria* sp. BL321 [27]. Five sesquiterpenes, including four oxygenated guaiane-type sesquiterpenes, xylaguaianols A−D (7**1**–**74**), and an iso-cadinane-type sesquiterpene isocadinanol A (7**5**), were isolated from the moss *Hypnum* sp.-derived fungus *Xylaria* sp. NC1214 [28]. Nine new oxygenated guaiane-type sesquiterpenes, including (1*S*,4*S*,5*R*,7*R*,10*R*,11*R*)-guaiane-5,10,11,12-tetraol (**76**), (1*S*,4*S*,5*R*,7*R*,10*R*,11*S*)-guaiane-1,10,11,12-tetraol (**77**), (1*S*,4*S*,5*R*,7*R*,10*R*,11*S*)-guaiane-5,10,11,12-tetraol (**78**), (1S,4*S*,5*S*,7*R*,10*R*,11*R*)-guaiane-1,10,11,12-tetraol (**79**), (1*R*,3*S*,4*R*,5*S*,7*R*,10*R*,11*S*)-guaiane-3,10,11,12-tetraol (**80**), (1*R*,3*R*,4*R*,5S,7*R*,10*R*,11*R*)-guaiane-3,10,11,12-tetraol (**81**), (1*R*,4*S*,5*S*,7*S*,9*R*,10S,11*R*)-guaiane-9,10,11,12-tetraol (**82**), (1*R*,4*S*,5*S*,7*R*,10*R*,11*S*)-guaiane-10,11,12-triol (**83**), and (1*R*,4*S*,5*S*,7*R*,10*R*,11*R*)-guaiane-10,11,12-triol (**84**) were isolated from the *Azadirachta indica*-derived fungus *Xylaria* sp. YM 311647. Compounds **76**–**84** were evaluated for their antifungal activities against *Candida albicans*, *Aspergillus niger*, *Pyricularia oryzae*, *Fusarium avenaceum*, and *Hormodendrum compactum*, with MIC values ranging from 32 to 512 μg/mL. Compounds **77** and **82** were the most potent ones against *C. albicans* with the same MIC value of 32 μg/mL. Compounds **77** and **79,** with the same substituted position of hydroxy groups, exhibited the most potent inhibitory activity against *A. niger* with the same MIC value of 64 μg/mL [29] (Figure 1).

#### 2.1.2. Diterpenes

Three new isopimarane diterpene derivatives, including xylongoic acids A–C (**85**–**87**), were isolated from the *Fomitopsis betulina*-derived fungus *Xylaria longipes* HFG1018 (Figure 2) [30]. One new diterpenoid, cubentriol (**88**), was isolated from the *L. akoensis* Hayata (Lauraceae)-derived fungus *Xylaria cubensis*. Two new compounds, including hypoxyterpoids A (**89**) and B (**90**), were separated from the mangrove *Bruguiera gymnorrhiza*-derived fungus *Hypoxylon* sp. (Hsl2–6). Compound **89** showed moderate *α*-glucosidase inhibitory activity (IC_50_, 741.5 *±* 2.83 µM) [31]. Compounds xylarianes A (**91**) and B (**92**) were obtained from *Xylaria* sp. 290, collected from Guizhou province, China [32]. Compounds spiropolin A (**93**) and myrocin E (**94**), with isopimarane-type skeletons, were isolated from the root of Mt. Gassan *Xylaria polymorpha*, Yamagata Prefecture, Japan [33]. Eighteen new diterpenes, including xylarinorditerpenes A–R (**95**–**112**) with nor-isopimarane skeletons, were isolated from the wood-rotting basidiomycete *Fomitopsis betulinus*-derived fungus *Xylaria longipes* HFG1018. Compounds **96**–**99**, **103**, and **108** showed immunosuppressive activity, with IC_50_ values varying from 1.0 to 51.8 μM [34]. Two new bioactive compounds, including acanthoic acid (**113**) and 3*β*,7*β*-dihydroxyacanthoic acid (**114**), were isolated from the fungus *Xylaria* sp. (EJCP07). Compound **114** demonstrated activity against *Bacillus subtilis*, with an MIC of 31.25 µg/mL. Also, **114** showed activity against *Escherichia coli*, with an MIC of 31.25 µg/mL. Both **113** and **114** exhibited the same MIC value of 62.5 µg/mL against *Salmonella typhimurium* [35]. Three new diterpene glycosides, including xylarcurcosides A–C (**115**–**117**) with isopimarane-type skeletons, were isolated from the *Alpinia zerumbet*-derived fungus *Xylaria* curta YSJ-5 [36]. Three new isopimarane diterpene glycosides, including 16-*α*-*D*-mannopyranosyloxyisopimar-7-en-19-oic acid (**118**), 15-hydroxy-16-*α*-*D*-mannopyranosyloxyisopimar-7-en-19-oic acid (**119**), and 16-*α*-*D*-glucopyranosyloxyisopimar-7-en-19-oic acid (**120**), were isolated from the fruit bodies of the fungus *Xylaria polymorpha*. Compounds **118**–**120** exhibited cytotoxicity against HL60, K562, HeLa, and lymph node carcinoma of the prostate (LNCaP) cell lines with IC_50_ values of 71–607 μM, respectively [37]. Two new isopimarane diterpenoids, including xylabisboeins A (**121**) and B (**122**), were isolated from the fungus *Xylaria* sp. SNB-GTC2501 [38]. Three new isopimarane diterpenes, including 14*α*,16-epoxy-18-norisopimar-7-en-4*α*-ol (**123**), 16-*O*-Sulfo-18-norisopimar-7-en-4*α*,16-diol (**124**), and 9-deoxy-hymatoxin A (**125**), were isolated from the *A. indica*-derived fungus *Xylaria* sp. YM 311647. Compound **124** exhibited inhibitory activity against *P. oryzae* with an MIC value of 32 μg/mL, while **125**, with a *γ*-lactone moiety and a sulfate group, showed the most potent activity against *C. albicans* and *P. oryzae*, with an MIC value of 16 μg/mL, and against *A. niger*, with an MIC value of 32 μg/mL, respectively [29]. Two novel diterpenes, including xylarilongipins A (**126**) and B (**127**) with an unusual cage-like bicyclo [2.2.2]octane moiety, were isolated from the medicinal plant *Fomitopsis betulinus*-derived fungus *Xylaria longipes* HFG1018. Compound **127** displayed moderate inhibitory activity against the cell proliferation of concanavalin A-induced T lymphocytes and lipopolysaccharide-induced B lymphocytes, with IC_50_ values of 13.6 and 22.4 μM, respectively [39] (Figure 2).

#### 2.1.3. Triterpenoid

Two new antibacterial terpenoids, including xylarioxides E–F (**128**–**129**), were isolated from the *Azadirachta indica*-derived fungus *Xylaria* sp. YM 311647 (Figure 3). Compound **128** displayed strongest inhibitory activity against *G. saubinetii*, *C. lunata*, and *C. gloeosporioides* (MIC, 8.0, 8.0, and 16.0 μg/mL). Compound **129** showed antibacterial activity against *A. alternata*, *C. lunata*, and *Colletotrichum gloeosporioides* (MIC, 8.0, 8.0, and 16.0 μg/mL) [20]. Compounds kolokosides A–D (**130**–**133**) were isolated from the Hawaiian wood-decay fungus *Xylaria* sp. Compound **130** exhibited antibacterial activity against *B. subtilis* and *S. aureus* at 200 µg/disk (inhibition zones: 16 and 12 mm, after 48 h, respectively) [40] (Figure 3).

### 2.2. Nitrogen-Containing Compounds

Nitrogen-containing compounds, including cytochalasan alkaloids and other nitrogen-containing metabolites, are notable for their exceptionally diverse class of secondary metabolites and potent bioactivities. A total of 112 new nitrogen-containing compounds were discovered from the genus *Xylaria* sp., including 67 cytochalasan alkaloids, and 45 other nitrogen-containing metabolites. Among them, 41 compounds showed cytotoxic activities, antibacterial activities, anti-inflammatory activities, enzyme-inhibitory activities, and other activities.

#### 2.2.1. Cytochalasan Alkaloids

Four new cytochalasans, including lagambasines A–D (**134**–**137**), were isolated from the *Palicourea elata*-derived fungus *Xylaria* sp. WH2D4 (Figure 4) [41]. One new compound karyochalasin A (**138**) was isolated from the fungus *X. karyophthora* [42]. Six new cytochalasins, including curtachalasins X1-X6 (**139**–**144**), were obtained from the plant *Solanum tuberosum*-derived fungus *Xylaria curta* E10. Compounds **139** and **143** showed cytotoxic activity against MCF-7 cell lines with IC_50_ values of 2.03 and 0.85 µM, respectively [43]. Two new cytochalasins, including 19,20-epoxycytochalasin Q (**145**) and deacetyl-19,20-epoxycytochalasin Q (**146**), were isolated from the wood-derived fungus *Xylaria obovate*. Compounds **145** and **146** displayed toxicity toward brine shrimp with the same LC_50_ values of 2.5 μg/mL, cytotoxic activity to HL-60 cell lines at the concentration of 1 μg/mL, and cytotoxicity against Vero cells with IC_50_ values of 0.46 and 1.9 μg/mL, respectively) [44]. Six new eytoehalasins, including 19,20-epoxycytochalasin R (**147**), 18-deoxy-19,20-epoxycytochalasin R (**148**), 18-deoxy-19,20-epoxycytochalasin Q (**149**), 19,20-epoxycytochalasin N (**150**), 19,20-epoxycytochalasin C (**151**), 21-acetylengleromycin (**152**) were isolated from the soil-derived fungus *Xylaria hypoxylon* [45]. Five new compounds 6,12-epoxycytochalasin D (**153**), 6-epi-cytochalasin P (**154**), 7-*O*-acetylcytochalasin P (**155**), 7-oxo-cytochalasin C (**156**), and 12-hydroxylcytochalasin Q (**157**), were isolated from the fungus *Xylaria longipes* (Ailao Moutain) [46]. One new cytochalasin, curtachalasin Q (**158**), was isolated from the fungus *Xylaria* sp. DO1801 [47]. Nine new epoxycytochalasans, including 19-epi-cytochalasin P1 (**159**), 6-epi-19,20-epoxycytochalasin P (**160**), 7-*O*-acetyl-6-epi-19,20-epoxycytochalasin P (**161**), 7-*O*-acetyl-19-epi-cytochalasin P1 (**162**), 6-*O*-acetyl-6-epi-19,20-epoxycytochalasin P (**163**), 7-*O*-acetyl-19,20-epoxycytochalasin C (**164**), 7-*O*-acetyl-19,20-epoxycytochalasin D (**165**), deacetyl-5,6-dihydro-7-oxo-19,20-epoxycytochalasin C (**166**), and 18-deoxy-21-oxo-deacetyl-19,20-epoxycytochalasin N (**167**), were isolated from the *Solanum tuberosum*-derived fungus *Xylaria* cf. *Curta*. Compounds **159**, **161**, and **165** showed strong cytotoxicity against HL-60 cell lines, with IC_50_ values of 13.31, 37.16, and 25.83 μM, respectively. Compound **162** showed potent inhibitory effects against MCF-7 cell lines with an IC_50_ value of 26.64 μM [48]. New compounds, including arbuschalasins A–D (**168**–**171**), were isolated from the *Bruguiera gymnorrhiza*-derived fungus *Xylaria arbuscula* GZS74 [49]. Two new open-chain cytochalasins, including xylarchalasins A and B (**172** and 1**73**), were isolated from the *Sophora tonkinensis*-derived fungus *Xylaria* sp. GDGJ-77B. Compound **173** displayed antibacterial activities against *B. subtilis* and *E. coli* with MIC values of 25 and 12.5 μg/mL, respectively [50]. Curtachalasins A (**174**) and B (**175**) were extracted from the stem of the *Solanum tuberosum*-derived fungus *Xylaria curta* (E10). Compounds (**174** and **175**) showed weak antibacterial activity against *M. gypseum* (70.3 and 68.4%, respectively, at the concentration of 200 μM) [51]. A new cytochalasin, cytochalasin P1 (**176**), was isolated from the marine-derived fungus *Xylaria* sp. SOF11 from the South China Sea. Compound **176** exhibited potent cytotoxicity against central nervous system carcinoma (SF-268) and MCF-7 cell lines with IC_50_ values of 1.37 and 0.71 μM, respectively [52]. Two new cytochalasins, including 18-deoxycytochalasin Q (**177**) and 21-*O*-deacetylcytochalasin Q (**178**), were isolated from the marine sediment-derived fungus *Xylaria* sp. SCSIO156. Compound **178** showed weak cytotoxicity against SF-268 and non-small cell lung cancer NCI-H460 cell lines, with MIC values of 44.3 and 96.4 μM, respectively [53]. A new cytochalasan alkaloid, xylastriasan A (**179**), with a rare 5/6/6/5/6 pentacyclic skeleton, was isolated from the fruiting bodies of the fungus *Xylaria striata*. Compound **179** showed weak cytotoxic activity against human hepatoma (HepG2), mouse melanoma (B16), and A549 cell lines with IC_50_ values of 93.61, 85.61, and 91.58 μM, respectively [54]. A new cytochalasin, cytochalasin H2 (**180**), obtained from the *Annona squamosa*-derived fungus *Xylaria* sp. (A23), exhibited weak cytotoxicity against HeLa and human non-hepatic 293T cells with 25.04 and 32.8% inhibition ration at the concentration of 1.0 μg/mL, respectively [55]. A halogenated hexacyclic cytochalasan, xylarichalasin A (**181**), with unprecedented 6/7/5/6/6/6 fused polycyclic skeletons, was obtained from the *Solanum tuberosum*-derived fungus *Xylaria* cf. *curta*. Compound **181** showed cytotoxicity against five human cancer cell lines including HL-60, A-549, human hepatocellular carcinoma (SMMC-7721), MCF-7, and human colon cancer (SW480) cells, with IC_50_ values of 17.3, 11.8, 8.6, 6.3, and 13.2 μM, respectively [56]. Two new cytochalasins, including cytochalasins D1 (**182**) and C1 (**183**) possessing a unique eleven-membered macrocycle with an oxygen bridge, were isolated from the *Solanum tuberosum*-derived fungus *Xylaria* cf. *curta*. Compounds **182** and **183** showed moderate cytotoxicity against human leukemia cell lines HL-60 with IC_50_ values of 12.7 and 22.3 μM, respectively [57]. Five new cytochalasans (**184**–**188**) were isolated from the fungus *Xylaria longipes* [46]. Eleven new cytochalasins, including curtachalasins F–P (**189**–**199**), were isolated from the *Solanum tuberosum*-derived fungus *Xylaria* cf. *curta*. The immunosuppressive assay against concanavalin A (ConA) induced T lymphocyte cell proliferation and lipopolysaccharide (LPS) induced B lymphocyte cell proliferation showed that **189** had significant selective inhibition on B-cell proliferation (IC_50_, 2.42 μM) and **198** had selective inhibition on T-cell proliferation (IC_50_, 12.15 μM). These results provide new clues to fulfill the urgent demand for new immunosuppressive drugs [58]. A new cytochalasin derivative, xylarisin B (**200**), was isolated from the mangrove-derived fungus *Xylaria* sp. HNWSW-2 [59] (Figure 4).

#### 2.2.2. Other Nitrogen-Containing Metabolites

One new alkaloid, akodionine (**201**), was isolated from the *L. akoensis* Hayata-derived fungus *Xylaria cubensis* (Figure 5) [11]. A new compound, xylactam B (**202**), was isolated from young healthy leaves of the *Tectaria zeylanica*-derived fungus *Xylaria* sp. [60]. A novel alkaloid, xylarialoid A (**203**), containing a [5,5,6] fused tricarbocyclic rings and a 13-membered macrocyclic moiety, was isolated from the leaves of the plant *Rauvolfia vomitoria*-derived fungus *Xylaria arbuscula*. Compound **203** exhibited potent cytotoxic activity against human A549 and HepG2 cell lines, with IC_50_ values of 14.6 and 15.3 µM, respectively. Also, **203** showed strong anti-inflammatory activity against LPS-induced nitric oxide (NO) production in RAW 264.7 cells, with an IC_50_ value of 6.6 µM [61]. One new compound, 2,3-dihydroxy-N-methoxy-6-propylbenzamide (**204**), was isolated from the *Hevea brasiliensis*-derived fungus *Xylaria* sp. PSU-H182 [62]. Xylopyridine A (**205**), isolated from the mangrove-derived fungus *Xylaria* sp., showed a strong DNA-binding affinity toward calf thymus (CT) DNA presumably via an intercalation mechanism [63]. A new compound, (*Z*)-3-{(3-acetyl-2-hydroxyphenyl) diazenyl}-2,4-dihydroxybenzaldehyde (**206**), was isolated from the lichen host *Amandinea medusulina*-derived fungus *Xylaria psidii*. Compound **206** showed moderate cytotoxicity against human lung cancer (NCI-H292) cell lines (IC_50_, 27.2 µg/mL) [64]. Xylanigripones A–C (**207**–**209**) were isolated from *Xylaria nigripes* (KL.) SACC. Compound **209** showed inhibitory activity against acetylcholinesterase (AChE) up to 38.1% at the concentration of 50 μM (positive control tacrine with 45.4% inhibition rate). Compound **209** exhibited inhibition of Cholesteryl Ester Transfer Protein activity with inhibition rates of 49% [65]. Xylariahgin F (**210**) was isolated from the *Isodon sculponeatus*-derived fungus *Xylaria* sp. [66]. Two new compounds, including (4*S*)-3,4-dihydro-4-(4-hydroxybenzyl)-3-oxo-1H-pyrrolo [2,1-*c*][1,4]oxazine-6-carbaldehyde (**211**) and methyl (2*S*)-2-[2-formyl-5-(hydroxymethyl)-1H-pyrrol-1-yl]-3-(4-hydr-oxyphenyl)propanate (**212**), were isolated from the Wuling Shen-derived fungus *Xylaria nigripes* [67]. A new cerebroside, allantoside (**213**), was isolated from *Xylaria allantoidea* SWUF76, and the fungus was collected from Phukhieo Wildlife Sanctuary [68]. Eight new compounds, including sinuxylamides A–D (**214**–**217**), assinuxylamide E (**218**), 4-(7,8-dihydroxy-4-oxoquinazolin-3(4H)-yl)butanoic acid (**219**), 4-(8-Hydroxy-4-oxoquinazolin-3(4H)-yl)butanoic acid (**220**), and 3,4-dihydroisocoumarin derivative 1′-N-Acetyl-5-methylmellein (**221**), were obtained from the *Sinularia densa*-derived fungus *Xylaria* sp. FM1005. Compounds **214** and **215** strongly inhibited the binding of fibrinogen to purified integrin IIIb/IIa in a dose-dependent manner, with IC_50_ values of 0.89 and 0.61 μM, respectively, and did not show cytotoxicity against human epithelial ovarian cancer A2780 and HEK 293 cells at 40 μM [69]. One new amide derivative, xylariamide (**222**), was isolated from the *Garcinia hombroniana*-derived fungus *Xylaria plebeja* PSU-G30 [70]. Compound xylaramide (**223**), isolated from the wood-inhabiting ascomycete *Xylaria longipes*, possessed potent antifungal activity against *Nematospora coryli* and *Saccharomyces cerevisiae*, with MIC values of 1.0 and 5.0 µg/mL, respectively [71]. Compound 2,5-diamino-N-(1-amino-1-imino-3-methylbutan-2-yl) pentanamide (**224**) was isolated from the fungus *Xylaria* cf. *cubensis* SWUF08–86 [72]. Compound xylariamino acid A, (**225**), a new amino acid derivative, was isolated from *Xylaria nigripes* (Kl.) Sacc. (Xylariaceae). The fungus was collected from Ailao Moutain, China [73]. Two new spirocyclic pyrrole alkaloids, including xylapyrrosides A (**226**) and B (**227**), were isolated from the Wuling Powder-derived fungus *Xylaria nigripes*. Compounds **226** and 2**27** were successfully synthesized, representing the first total synthesis of such spiroketal alkaloids with a pyranose ring. Compound **226** displayed antibacterial activity against *B. anthracis*, *B. megaterium*, *B. subtilis*, *S. aureus*, *E. coli*, *S. dysenteriae*, and S. *paratyphi*, with MIC values of 50, 25, 12.5, 25, 12.5, 25, and 25 μg/mL, respectively [74]. Two novel alkaloids, including (±)-xylaridines A (**228**) and B (**229**), were isolated from the genus *Xylaria longipes* Nitschke. Compound **228** possesses a 5/6/6/5/5 fused ring system with a unique 2-azaspiro [4.4]nonane substructure. Compound **228** showed weak antibacterial activity against *P. aeruginosa* with an MIC value of 128 μg/mL, while **229** displayed activity against *S. entericawith* with an MIC value of 93 μg/mL [75]. One new compound, (−)-xylariamide A (**230**), was isolated from the outer bark of the *Glochidion ferdinandi*-derived fungus *Xylaria* sp. Compound **230** displayed toxicity against brine shrimp (*Artemia salina*) with 0% and 71% lethality at the concentrations of 20 and 200 μg/mL, respectively [76].

Cyclotripeptide X-13 (**231**) and its derivatives xyloallenoide A (**232**), xyloallenoide A1 (**233**), and cyclotripeptide X-13a (**234**), were isolated from the mangrove-derived fungus *Xylaria* sp. (No. 2508). Compound 2**32** and its diastereomer **233** were totally synthesized. Compound **231** and its derivatives **232**–**234** concentration-dependently promoted angiogenesis in zebrafish in vivo and endothelial cell cultures in vitro. Compound **231** dose-dependently induced angiogenesis in zebrafish embryos and human endothelial cells, indicating that **231** possesses potent angiogenic properties that are promising for development as a novel class of pro-angiogenic agents for angiotherapy [77,78]. Xylaroamide A (**235**), isolated from *Xylaria* sp. 218–066, exhibited cytotoxic activity against human basal-like breast cancer (BT-549) and human colon cancer (RKO) cell lines with IC_50_ values of 2.5 and 9.5 μM, respectively. This fungus was isolated from a sample of *Usnea* sp. collected from Linzhi, Tibet, China [79]. Two new cyclopeptides, including xylarotides A (**236**) and B (**237**), were isolated from *Xylaria* sp. 101. The fungus was collected from the fruiting body of *Xylaria* sp. collected from Gaoligong Mountain, China [80]. Two new cyclopentapeptides, including xylapeptide A (**238**) with an uncommon *L*-pipecolinic acid moiety and xylapeptide B (**239**), were isolated from the *Sophora tonkinensisan*-derived fungus *Xylaria* sp. GDG-102. Compounds **238** and **239** were totally synthesized, and **238** showed moderate activity against *B. cereus* and *B. subtilis*, with the same MIC value of 125 µg/mL [81]. Three new proline-containing cyclic nonribosomal peptides, including ellisiiamides A–C (**240**–**242**), were isolated from *the* blueberry *Vaccinium angustifolium*-derived fungus *Xylaria ellisii*. Compound **240** showed modest inhibitory activity against *E. coli*, with an MIC value of 100 μg/mL [82]. Two new cyclic pentapeptides, including cyclo(*N*-methyl-*L*-Phe-*L*-Val-*D*-Ile-*L*-Leu*-L*-Pro) (**243**) and cyclo(*L*-Val-*D*-Ile-*L*-Leu-*L*-pro-*D*-Leu) (**244**), were isolated from the lichen *Leptogium saturninum*-derived fungus *Xylaria* sp. Compound **243** showed synergistic antifungal activity against *C, albicans* SC5314 with an MIC value of 0.004 μg/mL [83]. A new cyclic pentapeptide, pentaminolarin (**245**), was isolated from the wood-decaying fungus *Xylaria* sp. (SWUF08–37). Compound **245** showed weak cytotoxic activity against Vero, HeLa, HT29, HCT116, and MCF-7 cell lines, with IC_50_ values of 67.89, 44.98, 31.92, 37.98, and 14.62 μg/mL, respectively [84] (Figure 5).

### 2.3. Polyketides

Polyketides are a class of compounds characterized by their exceptionally diverse structures and bioactivities. Polyketides are generated through a series of Claisen condensation reactions involving acetyl-CoA, malonyl-CoA, and so on. A total of 70 new polyketides were discovered from the genus of *Xylaria* sp., and 23 of them had cytotoxic activities, antibacterial activities, anti-inflammatory activities, enzyme-inhibitory activities, and so on.

Two new cyclohexenones, including xylariacyclones A (**246**) and B (**247**), were isolated from the *Garcinia hombroniana*-derived fungus *Xylaria plebeja* PSU-G30 (Figure 6) [74]. A new compound, xylarianin B (**248**), was isolated from the *Panax notoginseng*-derived fungus *Xylaria* sp. SYPF 8246 [85]. One new compound, xylariaone (**249**), was isolated from the fungal strain *Xylaria* sp. 12F075 [86]. A pair of new chromone derivatives, including (+)-xylarichromone A (**250**) and (−)-xylarichromone A (**251**), were isolated from the fungus *Xylaria nigripes* (Ailao Moutain, China). The neuroprotective effects of **250** against oxygen and glucose deprivation (OGD)-induced pheochromocytoma-12 cell (PC12) injury were tested, and it was found that **255** significantly enhanced PC12 cell line viability and inhibited apoptosis at the concentrations of 0.1 and 1 µM [87]. Five new 2,5-diarylcyclopentenones, including xylariaones A1-B2 (**252**–**255**) and xylaripyone H (**256**), were isolated from the *Cudrania tricuspidata*-derived fungus *Xylaria* sp. [88]. One new azaphilone derivative, xylariphilone (**257**), was isolated from the seagrass *Halophila ovalis*-derived fungus *Xylaria* sp. PSU-ES163 [89]. Three new dimeric chromanones, including xylaromanones A–C (**258**–**260**), and one new cyclohexenone, (*R*)-4-Hydroxy-2-ethyl-2-cyclohexen-1-one (**261**), were isolated from the lamina of the *Hevea*. *Brasiliensis*-derived fungus *Xylaria* sp. PSU-H182 [62]. A new tetralone derivative, 3,4,5-trihydroxy-1-tetralone (**262**), was isolated from termite nest-derived fungus *Xylaria* sp. [14]. One new compound, hemi-cycline A (**263**), was isolated from the fungus *Xylaria* cf. *cubensis* SWUF08-86 (Phu Khieo Wildlife Sanctuary, Thailand) [76]. A new 2H-chromene derivative, hexacycloxylariolone (**264**), isolated from the plant-associated fungus *Xylaria* sp., showed inhibitory effects on the growth of THP-1 cells with an IC_50_ value of 82.3 µg/mL [90]. Two new *γ*-pyrones, including xylaropyrones B (**265**) and C (**266**), were isolated from the *Spartina maritima*-derived fungus *Xylaria* sp. SC1440 [91]. Two new benzoquinone metabolites, including 2-chloro-5-methoxy-3-methylcyclohexa-2,5-diene-1,4-dione (**267**) and xylariaquinone A (**268**), were isolated from the *Sandoricum koetjape*-derived fungus *Xylaria* sp. Compounds **266** and **267** showed activity against *P. falciparum*, K1 strain, with IC_50_ values of 1.84 and 6.68 µM, respectively. Compound **266** also showed cytotoxicity against Vero cells with an IC_50_ value of 1.35 µM [92]. Ten new compounds, including xylanthraquinone (**269**), xyloketals A–H (**270**–**277**), and xyloketal J (**278**), were isolated from the mangrove-derived fungus *Xylaria* sp. (No. 2508). Compounds **270**–**278** shared identical 5,6-bicyclic acetal moieties fused to a benzene ring in the center. Compounds **270**–**278** were able to act in a number of different disease models due to the similarity in the underlying pathological mechanisms, including oxidative stress, NO disturbance, intracellular Ca^2+^ imbalance, and protein aggregation. Compound **271** also showed alleviation of lipid accumulation in a non-alcoholic fatty liver disease model, and treatment with this compound also induces glioblastoma cell death [93,94,95,96,97,98]. Eleven new chromanones, including paecilins F–P (**279**–**289**), were isolated from the potato tissue-derived fungus *Xylaria curta* E10. Compounds **285** and **287** showed antibacterial activity against *E. coli* with the same MIC value of 16 µg/mL [99]. Three new azaphilone derivatives, including rubiginosins A–C (**290**–**292**), were isolated from the *Fraxinus excelsior*-derived fungus *Xylariaceus ascomycete* [100]. One new compound, xylaphenoside A (**293**), was obtained from the *Selaginella moellendorffii*-derived fungus *Xylaria* sp. CGMCC No. 5410 and showed antimicrobial activity against *S. aureus*, with an IC_50_ value of 6.2 µg/mL [101]. Three cyclohexenoneesordaricin derivatives, including xylarinonericins A–C (**294**–**296**), were isolated from the *G. hombroniana*-derived fungus *Xylaria plebeja* PSU-G30 [71]. Three new azaphilone derivatives, including rubiginosins A–C (**297**–**299**), were isolated from the fruit bodies of *Xylariaceus ascomycete* [102]. Three new polyketides, including 1,3,8-Trihydroxy-7-methoxy-9-methyldibenzofuran (**300**) (6*S*,2′*R*,6′*S*)-6-Methyl-2-((6-methyltetrahydro-2H-pyran-2-yl)methyl)-2,3-dihydro-4H-pyran-4-one (**301**), and (2′*R*,6′*S*)-5-((-6-Methyltetrahydro-2H-pyran-2-yl)methyl)benzene-1,3-diol (**302**), were isolated from the *Geophila repens*-derived fungus *Xylaria feejeensis*. Compound **300** showed cytotoxic activity against Vero cells and the HCT116, HT29, MCF-7, and HeLa cell lines with IC_50_ values of 25.00, 14.36, 8.99, 18.40, and 16.68 µg/mL, respectively. Compound **302** showed cytotoxic activity against HCT116, HT29, MCF-7, and HeLa with IC_50_ values of 17.50, 19.49, 50.24, and 13.53 µg/mL [103]. Two new compounds, including 6′,7′-didehydrointegric acid (**303**) and 13-carboxyintegric acid (**304**), were isolated from the *Geophila repens*-derived fungus *Xylaria feejeensis* [104]. One new compound (4*S*,5*S*,6*S*)-5,6-epoxy-4-hydroxy-3-methoxy-5-methyl-cyclohex-2-en-1-one (**305**), was isolated from the fungus *Xylaria carpophila*, which was collected from Gaoligong Mountains in Yunnan Province. Compound **305** showed significant specific cytotoxicity against the HL-60, A-549 MCF-7, and SW480 cell lines, with IC_50_ values of 23.1, 35.7, 28.5, and 29.0 μM, respectively [19]. Two new tetralone derivatives, including xylariol A (**306**) and B (**307**), were isolated from the fresh stems of the *Ligustrum lucidum*-derived fungus *Xylaria hypoxylon* AT-028. Compounds **306** and **307** showed moderate cytotoxic activities against HepG2 cells, with IC_50_ values of 22.3 and 21.2 μg/mL, respectively [105]. One new polyketide, 1-(xylarenone A)xylariate A (**308**), was isolated from the medicinal *Torreya jackii*-derived fungus *Xylaria* sp. NCY2 [106]. Two new polyketides, including schweinitzins A (**309**) and B (**310**), were isolated from the fungus *Xylaria schweinitzii* Berk. and M.A. Curtis. Compound **309** exhibited cytotoxicity against KB, Hep-G2, human lung adenocarcinoma (SK-Lu-1), and MCF-7 cell lines with IC_50_ values of 0.72, 2.13, 2.32, and 4.09 μg/mL, respectively [107]. Two new polyketides, including 6-ethyl-8-hydroxy-4H-chromen-4-one (**311**) and 6-ethyl-7,8-dihydroxy-4H-chromen-4-one (**312**), were isolated from the fungus *Xylaria* sp. SWUF09-62. Compound **312** exhibited anti-inflammatory properties by reducing nitric oxide production in LPS-stimulated RAW264.7 cells, with an IC_50_ value of 1.57 ± 0.25 μg/mL, and cytotoxicity against HT29 cells, with an IC_50_ value of 16.46 ± 0.48 μg/mL [108]. A novel polyketide, mellisol (**313**), was isolated from the fungus *Xylaria mellisii* (BCC 1005). Compound **313** exhibited antivirus activity against herpes simplex virus type 1, with an IC_50_ value of 10.50 μg/mL, and also showed cytotoxic activity against Vero cells with an IC_50_ value of 45.8 μg/mL [109]. A new compound, *γ*-pyrone-3-acetic acid (**314**), was obtained from a bark sample of a live oak-derived fungus *Xylatia* sp. [110]. One new *α*-pyrone, 9-hydroxyxylarone (315), was isolated from the moss *Hypnum* sp.-derived fungus *Xylaria* sp. NC1214 [28] (Figure 6).

### 2.4. Lactones

Lactones represent a class of compounds that contain lactone rings within their molecular structure. A total of 76 new lactones were discovered from the genus *Xylaria* sp. Remarkably, 32 compounds had cytotoxic activities, antioxidant activities, antibacterial activities, anti-inflammatory activities, enzyme-inhibitory activities, and so on.

One new compound, xylarolide (**316**), was isolated from the fungus *Xylaria* sp. 101, which was isolated from the fruiting body of *Xylaria* sp. collected from Gaoligong Mountain, China (Figure 7) [80]. One new compound, (3*αS*,6*αR*)-4,5-dimethyl-3,3*α*,6,6*β*-tetrahydro-2Hcyclopenta [*β*]furan-2-one (**317**), isolated from the fungus *Xylaria curta* 92092022, showed moderate antibacterial activity against *S. aureus* NBRC 13276 with the inhibition zone of 12 mm at a concentration of 100 μg/disk [111]. A new compound, xylarphthalide A (**318**), was isolated from the *Sophora tonkinensis*-derived fungus *Xylaria* sp. GDG-102. Compound **318** showed antibacterial activities against *Bacillus megaterium*, *B. subtilis*, *S. aureus*, *E. coli*, and *Shigella dysenteriae*, with MIC values of 12.5–25 μg/mL [112]. Two new dihydroisocoumarin glycosides, including xylarglycosides A (**319**) and B (**320**), were isolated from the *Illigera celebica*-derived fungus *Xylaria* sp. KYJ-15 and showed antibacterial activities against *S. aureus* with MIC values of 4 and 2 μg/mL, respectively. These compounds also exhibited 2,2-diphenyl-1-(2,4,6-trinitrophenyl) hydrazyl stable free radical (DPPH) radical scavenging activity, with IC_50_ values of 9.2 ± 0.03 and 13.3 ± 0.01 μM, respectively [113]. One new isocoumarin, akolitserin (**321**), was isolated from the *L. akoensis*-derived fungus *Xylaria cubensis* [11]. One new compound, 5-*O*-*α*-Dglucopyranosyl-5-hydroxymellein (**322**), obtained from the *Selaginella moellendorffii*-derived fungus *Xylaria* sp. CGMCC No. 5410, showed antimicrobial activity against *S. aureus* and *Micrococcus luteus* with IC_50_ values of 6.2 and 6.2 µg/mL, respectively [101]. Seven new compounds, including xylaripyone A–G (**323**–**329**), were isolated from the *Cudrania tricuspidata*-derived fungus *Xylaria* sp. Compound **326** showed moderate cytotoxic activity against PC3 cell lines, with an IC_50_ value of 14.75 μM. Compound **328** displayed weak inhibitory activity against NO production in RAW 264.7 murine macrophages, with IC_50_ values of 49.76 and 69.68 *μ*M, respectively [98]. Four new isocoumarins, including hypoxymarins A–D (**330**–**333**), were obtained from the mangrove *Bruguiera gymnorrhiza*-derived fungus *Hypoxylon* sp. (Hsl2-6). Compound **332** showed DPPH radical scavenging activity with an IC_50_ value of 15.36 *±* 0.24 µM, which was better than the positive control ascorbic acid (IC_50_, 20.49 *±* 0.43 µM) [30]. A new lignanoid, xylarianolide (**334**), was isolated from the fungal strain *Xylaria* sp. [114]. Six new compounds, including xylariahgins A–F (**335**–**340**) and 3-(2,3-dihydroxypropyl)-6,8-dimethoxyiso coumarin (**341**), were isolated from the *Isodon sculponeatus*-derived fungus *Xylaria* sp. hg1009 [66]. Four new *α*-pyrones, including xylariaopyrones A–D (**342**–**345**), were isolated from the fungus *Xylariales* sp. (HM-1). Compounds **342** and **343** were a pair of epimers with a ketal function group. Compounds **342**–**345** displayed inhibiting activity against plant-pathogeni*c Erwinia carotovora*, with MIC values ranging from 25.5 to 20.5, 22.6, and 24.7 μg/mL, respectively. Compounds **342**–**345** also showed brine shrimp inhibiting activity with inhibiting ratios of 42%, 76%, 84%, and 82%, respectively, at a concentration of 50 μg/mL [115]. A new compound, 3*S*-hydroxy-7melleine (**346**), was isolated from the mangrove-derived fungus *Xylaria* sp. (No. 2508) [116]. Two new compounds, including pestalotin 4′-*O*-methyl-*β*-mannopyranoside (**347**) and 3*S*,4*R*-(+)-4-hydroxymellein (**348**), were isolated from the *Hintonia latiflora*-derived fungus *Xylaria* feejeensis. Compound **348** inhibited *Saccharomyces cerevisiae α*-glucosidase (*α*GHY) with an IC_50_ value of 441 ± 23 μM, which was better than the positive control acarbose (IC_50,_ 545 ± 19 μM) [117]. Two new compounds, including xylarellein (**349**) and xylariaindanone (**350**), were isolated from the *Garcinia hombroniana*-derived fungus *Xylaria* sp. PSU-G12 [118]. Six new *α*-pyrones, including xylapyrones A-F (**351**–**356**), were isolated from the *Saccharum arundinaceum*-derived fungus *Xylaria* sp. BM9 [119]. One new compound, 6-heptanoyl-4-methoxy-2H-pyran-2-one (**357**), isolated from the leaf of the *Sophora tonkinensis*-derived fungus *Xylaria* sp. GDG-102, showed antimicrobial activity against *E. coli* and *S. aureus*, with the same MIC value of 50 μg/mL [120]. Two lactones, including (+)-phomalactone (**358**) and 6-(1-propenyl)-3,4,5,6-tetrahydro-5-hydroxy-4Hpyran-2-one (**359**), were isolated from the *Siparuna* sp.-derived fungus *Xylaria* sp. Grev. (Xylariaceae). Compound **358** showed weak anti-plasmodial activity against *P. falciparum*, with an IC_50_ value of 13 µg/mL [121]. One new compound, xylaolide A (**360**), was isolated from the mangrove sediment-derived fungus *Xylariaceae* sp. DPZ-SY43, which was collected in Sanya [122]. Two new lactones, including (*S*)-8-Hydroxy-6-methoxy-4,5-dimethyl-3-methyleneisochroman-1-one (**361**) and (*R*)-7-hydroxy-3-((*R*)-1-hydroxyethyl)-5-methoxy-3,4-dimethylisobenzofuran-1(3H)-one (**362**), were isolated from the mangrove-derived fungus *Xylaria* sp. BL321 [123]. Five new *α*-pyrones, including xylariaopyrones E–I (**363**–**367**), were isolated from the fungus *Xylaria* sp. (HM-1). Compound **363** is the first example of an α-pyrone derivative with a novel [3, 2, 0] bridge ring system via a ketal function group in the side chain. Compounds **363**–**365** showed moderate inhibiting activities against *E. coli*, *S. aureus*, and *P. aeruginosa* with MIC values from 25.4 to 64.5 μg/mL, and compound **367** showed significant inhibition of monoamine oxidase B with an IC_50_ value of 15.6 μM [124]. Two new polyketides, including lasobutones A–B (**368**–**369**), were isolated from the *Coptis chinensis*-derived fungi *Xylaria* sp. Compound **369** showed inhibitory activity against the NO production in the LPS-induced macrophage RAW264.7, with an IC_50_ value of 42.5 μM [125]. One new compound, coloratin A (**370**), isolated from the Xylariaceous mushroom *Xylaria intracolorata*, showed strong antimicrobial activity against *Klebsiella pneumoniae* with inhibition zones of 22 mm at a dose of 50 mg per paper disk [126]. Two new compounds, including (3*R*)-6-methoxy-5-methoxycarbonylmellein (**371**) and (3*S*,2′*R*,6′*R*)-asperentin-8-*O*-methylether (**372**), were obtained from the medicinal plant *Geophila repensfungu*-derived fungus *Xylaria feejeensis.* Compound **371** showed cytotoxic activity against HCT116 and HT29 cell lines, with IC_50_ values of 92.93 and 96.42 µg/mL, respectively. Compound **372** showed cytotoxic activity against HCT116, HT29, and HeLa cell lines with IC_50_ values of 45.09, 67.60, and 92.93 µg/mL, respectively [93]. A new nonenolide, xyolide (**373**), was isolated from *Xylaria feejeensis* (Berk.) Fr. (E6912B), which displayed antifungal activity against the plant pathogen *P. ultimum* with an MIC value of 425 μM [127]. Two new 10-membered lactones, including multiplolides A (**374**) and B (**375**), were isolated from the fungus *Xylaria multiplex* BCC 1111. Compounds **374** and **375** exhibited antifungal activity against *C. albicans* with IC_50_ values of 7 and 2 μg/mL, respectively [128]. Four new latones, including xylariolide A (**376**), xylariolide B (**377**), xylariolide C (**378**), and xylariolide D (**379**), were isolated from the medicinal plant *Torreya jackii*-derived fungus *Xylaria* sp. NCY2 [106]. One new compound, furofurandione (**380**), was purified from the plant palm *Licuala spinose*-derived fungus *Xylaria* sp. (BCC 21097) [24]. One new dihydroisocoumarin, (3*R*,4*R*)-3,4-dihydro-4,6-dihydroxy-3-methyl-1-oxo-1H-isochromene-5-carboxylic acid (**381**), was obtained from the plant *Piper aduncum*-derived fungus *Xylaria* sp. Compound **381** exhibited antifungal activity against *C. cladosporioides* and *C. sphaerospermum* with detection limits of 10.0 and 25.0 μg, respectively. Compound **381** also exhibited AChE inhibitory activity with a detection limit of 3.0 μg [129]. Two new compounds, including (3*S*)-3,4-dihydro-8-hydroxy-7-methoxy-3-methylisocoumarin (**382**) and (3*S*)-3,4-dihydro-5,7,8-trihydroxy-3-methylisocoumarin (**382**), were isolated from the fungus *Xylaria* sp. SWUF09-62. Compound **383** exhibited anti-inflammatory activity by reducing NO production in LPS-stimulated RAW264.7 cells (IC_50_, 3.02 ± 0.27 μg/mL) and cytotoxicity against HT29 cells (IC_50_, 97.78 ± 7.14 μg/mL) [108]. Two new compounds, including pestalotin 4’-*O*-methyl-*β*-mannopyranoside (**384**) and 3*S*,4*R*-(+)-4-hydroxymellein (**385**), were isolated from the plant *Hintonia latiflora*-derived fungus *Xylaria feejeensis*. Compound **385** showed inhibition activity against *Saccharomyces cerevisiae* α-glucosidase (αGHY) with an IC_50_ of 441 ± 23 μM, which was better than the positive control acarbose (IC_50_, 545 ± 19 μM). Molecular docking predicted that **385** bound to αGHY in a site different from the catalytic domain, which could imply an allosteric type of inhibition [117]. A new phytotoxic bicyclic lactone, (3a*S*,6a*R*)-4,5-dimethyl-3,3a,6,6a-tetrahydro-2H-cyclopenta [*β*]furan-2-one (**386**), was isolated from the fungus *Xylaria curta* 92092022. Compound **386** showed moderate antibacterial activity against both *Pseudomonas aeruginosa* ATCC and *S. aureus* NBRC, with the same inhibition zone of 13 mm at a concentration of 100 μg/disk [111]. A new ten-membered macrolide (**387**) and a new *α*-pyrone derivative (−)-annularin C (**388**) were isolated from the marine sponge *Stylissa massa*-derived fungus *Xylaria feejeensis*. Compound **388** exhibited significant down-regulating activity of osteoclast cell differentiation at concentrations of 0.5 and 1 μM [130]. Two new alpha-pyrone derivatives, including xylarone (**389**) and 8,9-dehydroxylarone (**390**), were isolated from the wood-derived fungus *Xylaria hypoxylon* A27-94. Compound **389** displayed anti-proliferative activity against human colon adenocarcinoma (Colo320) and mouse leukemic (L1210) cell lines, with IC_50_ values of 40 and 50 μg/mL, respectively. Compound **390** displayed anti-proliferative activity against the Colo-320, L1210, and HL-60 cell lines, with IC_50_ values of 25, 25, and 50 μg/mL, respectively [131] (Figure 7).

### 2.5. Other Classes

There were also some other classes of secondary metabolites isolated from *Xylaria* spp., such as fatty acids, steroids, and benzene derivatives. A total of 54 new compounds were isolated from the genus of *Xylaria* sp., and 26 of them showed cytotoxic activities, antibacterial activities, anti-inflammatory activities, enzyme-inhibitory activities, and so on.

A new fatty acid, rubiginosic acid (**391**), was isolated from the fruit bodies of the *Corylus avellana*-derived fungus *Xylariaceus ascomycete* (Figure 8) [102]. Three new compounds, including xylarianin A (**392**), xylarianin C (**393**), and xylarianin D (**394**), and three new natural products, including 6-methoxycarbonyl-2′-methyl-3,5,4′,6′-tetramethoxy-diphenyl ether (**395**), 2-chlor-6-methoxycarbonyl-2′-rnethyl-3,5,4′,6′- tetramethoxy-diphenyl ether (**396**), and 2-chlor-4′-hydroxy-6-methoxy carbonyl-2′-methyl-3,5,6′-trimethoxy-diphenyl ether (**397**), were isolated from the *Panax notoginseng*-derived fungus *Xylaria* sp. SYPF 8246. Compounds **392** and **395**–**397** displayed significant inhibitory activities against Human Carboxylesterase 2 (hCE-2), with IC_50_ values of 10.43, 6.69, 12.36, and 18.25 µM, respectively [85]. Two new steroids, including xylarsteroids A (**398**) and B (**399**), the first examples of the C28-steroid with an unusual *β*- and *γ*-lactone ring, were isolated from the *Illigera celebica*-derived fungus *Xylaria* sp. KYJ-15. Compound **398** exhibited potent AChE inhibitory activity, with an IC_50_ value of 2.61 ± 0.05 μM. Compounds **398** and **399** exhibited strong antibacterial activity against *B. subtilis*, with the same MIC value of 2 μg/mL [114]. One aliphatic derivative, akoenic acid (**400**), was isolated from leaves of the *L. akoensis* Hayata (Lauraceae)-derived fungus *Xylaria* cubensis [8]. One new isovaleric acid, phenethylester (**401**), isolated the fungus *Xylaria nigripes* (Kl.) Sacc. (Xylariaceae), significantly reduced the percentage of apoptotic cells at a concentration of 1 µM [73]. One new metabolite, 3,7-dimethyl-9-(-2,2,5,5-tetramethyl-1,3-dioxolan-4-yl) nona-1,6-dien-3-ol (**402**), was isolated from a *Taxus mairei*-derived strain. Compound **402** exhibited antibacterial activity against *B. subtilis* ATCC 9372 48.1%, *B. pumilus* 7061 31.6%, and *S. aureus* ATCC 25923 47.1%, at a concentration of 10 μg/mL [132]. A new fatty acid, rubiginosic acid (**403**), was obtained from the *Fraxinus excelsior*-derived fungus *X. ascomycete* [100]. Two new glucosides, including xylarosides A (**404**) and B (**405**), were isolated from the *Garcinia dulcis*-derived fungus *Xylaria* sp. PSU-D14 [133]. One new phenyloxolane compound, 2-methyl-2-(4-hydroxymethylphenyl) oxacyclopentane (**406**), was isolated from the fungus *Xylaria polymorpha* (Pers.: Fr.) Grer. Compound **406** showed moderate inhibitory activity against *Panagrellus redivivus*, with a mortality ratio of 59.6% at 2.5 mg/mL [13]. Four new alkyl aromatics, including penixylarins A–D (**407**–**410**), were isolated from a mixed culture of the fungus *Penicillium crustosum* PRB-2 and the mangrove-derived fungus *Xylaria* sp. HDN13-249. Compounds **408** and **409** showed inhibitory activity against *M. phlei*, *B. subtilis*, and *V. parahemolyticus*, with MIC values ranging from 6.25 to 100 μM, and compound **409** also possessed potential anti-tuberculosis effects against *Mycobacterium phlei*, with an MIC value of 6.25 μM [134]. One new phenylacetic acid derivative (**411**) and one new naphthalenedicarboxylic acid (**412**) were isolated from the *Sinularia densa*-derived fungus *Xylaria* sp. FM1005 [69]. A new polyalcohol xylatriol (**413**) was isolated from the plant-associated fungus *Xylaria* sp. [89]. Three new benzofurans, including acumifurans A–C (**414**–**416**), were isolated from the nests of the *Odontotermes formosanus*-derived fungus *X. acuminatilongissima* YMJ623 [135]. A new fatty acid, (2*E*,4*E*,6*S*)-6-hydroxydeca-2,4-dienoic acid (**417**), was isolated from the gorgonian-derived fungus *Xylaria* sp. C-2, which was collected from the South China Sea [136]. Two new steroids, including (24*R*)-22,23-dihydroxy-ergosta-4,6,8(14)-trien-3-one 23-*β*-*D*-glucopyranoside (**418**) and xylarester (**419**), were isolated from the fungus *Xylaria* sp. Compound **418** showed cytotoxicity against MCF-7 cell lines with a ratio of inhibition at 72% for a concentration of 40 μM [137]. One new compound, coloratin B (**420**), isolated from the Xylariaceous mushroom *X. intracolorata*, showed strong antimicrobial activity against *K. pneumoniae*, with inhibition zones of 22 mm at a dose of 50 mg per paper disk [138]. Three new methylsuccinic acid derivatives, including xylaril acids A–C (**421**–**423**), and two enoic acid derivatives, including xylaril acids D and E (**424** and **425**), were isolated from the fungus *Xylaria longipes*. Compounds **421**–**425** showed no toxic effects on PC12 cells at a concentration of 10 μM. Compounds **421**–**425** displayed neuroprotective activities against OGD/R injury in PC12 cells by enhancing cell viability and inhibiting cell apoptosis [99]. A new compound, wheldone (**426**), was isolated from the coculture of *Aspergillus fischeri* (NRRL 181) and *Xylaria flabelliformis* (G536). Compound **426** displayed cytotoxic activity against breast cancer MDA-MB-231, ovarian carcinoma OVCAR-3, and breast carcinoma MDA-MB-435 cell lines, with IC_50_ values of 7.6, 3.8, and 2.4 μM, respectively [139]. Seven new isoprenyl phenolic ethers, including fimbriethers A–G (**427**–**433**), were isolated from termite nest-derived fungus *Xylaria fimbriata* Lloyd (YMJ491). Compound **433** exhibited the strongest NO inhibition activity with an average maximum inhibition (Emax) of 49.7% at the concentration of 100 μM. Compounds **428** and **431** showed moderate iNOS inhibitory activity with Emax values of 31.3 and 38.9%, respectively. Research on the structure–activity relationship indicated that the methyl benzoate moiety was a possible active site [140]. Two new compounds, including xylarioic acid B (**434**) and xylariate C (**435**), were isolated from the medicinal plant *Torreya jackii*-derived fungus *Xylaria* sp. NCY2 [106]. A novel 20-norpimarane glucoside, xylopimarane (**436**), isolated from the fungus *Xylaria* sp. (BCC 4297), displayed cytotoxic activity against the KB, MCF-7, and NCI-H187 cell lines, with IC_50_ values of 1.0, 13, and 65 μM, respectively [141]. Two new compounds, including xylarinic acids A (**437**) and B (**438**), were isolated from the fruit body of *Xylaria polymorpha* (Pers.) Grev. Compounds **437** and **438** showed strong antifungal activity against *P. ultinum* and *M. grisea* with an inhibition zone of 16–20 mm diameter. They also showed antifungal activity against *A. panax*, *A. niger*, and *F. oxysporium* [142]. Two new succinic acid derivatives, including xylacinic acids A (**439**) and B (**440**), were isolated from the mangrove-derived fungus *Xylaria cubensis* PSU-MA34 [143]. A new cerebroside, allantoside (**441**), was isolated from the fungus *Xylaria allantoidea* SWUF76. The fungus was collected from Phukhieo Wildlife Sanctuary, Thailand [68]. A new fluorescent compound, ergosta-4,6,8(14),22-tetraen-3-one (**442**), was isolated from the fungus *Xylaria* sp., which was collected in Vietnam. Compound **442** showed inhibitory activity of NO production in RAW 264.7 cells stimulated by lipopolysaccharide, with an IC_50_ value of 28.96 µM [144]. Two new glucoside derivatives, including xylarosides A (**443**) and B (**444**), were isolated from the leaves of the *Garcinia dulcis*-derived fungus *Xylaria* sp. PSU-D14 [133]. A new compound, methyl aminobenzoate (**445**), was isolated from the wood-decayed fungus *Xylaria* sp. BCC 9653 [145].

## 3. Comprehensive Overview and Conclusions

In this review, the sources, structural diversity, and biological activity of secondary metabolites from *Xylaria* fungi are summarized, covering the period from 1994 to January 2024. A total of 445 new compounds were obtained from the genus *Xylaria*. A sample of 177 notable compounds and their biological activities are summarized in Table 1. The structural diversities and bioactivities of the new secondary metabolites discovered from *Xylaria* spp. are also shown in Figure 9.

The chemical structures of the 445 new secondary metabolites from *Xylaria* fungi can mainly be classified into five types, including 133 terpenoids, 112 nitrogen-containing compounds, 70 polyketones, 76 lactones, and 54 other compounds consisting of steroids, fatty acids, and benzene derivatives (Figure 9). However, among these 445 new compounds, terpenoids predominantly accounted for 29.89%, while nitrogen-containing compounds, polyketides, lactones, and other types accounted for 25.18%, 15.73%, 17.07%, and 12.13, respectively.

Moreover, it is worth noting that nearly 39.8% (177 compounds) showed broad-spectrum biological activities, including cytotoxic (52 compounds), antimicrobial (38 compounds), antifungal (30 compounds), anti-inflammatory (18 compounds), enzyme inhibition (12 compounds), immunosuppressive (10 compounds), and other activities (17 compounds). Notably, cytotoxic (29.37%), antibacterial (21.46%), and antifungal (16.95%) activities represent the top three bioactivities (Figure 9). It is important to highlight that many compounds exhibit multiple activities. For example, xyloketal B (**271**) is able to act in a number of different disease models in the underlying pathological mechanisms, including oxidative stress, NO disturbance, intracellular Ca^2+^ imbalance, and protein aggregation.

In summary, *Xylaria* fungi have been proven to be an important source of structured novel and diverse secondary metabolites with a broad range of biological activities, revealing their great untapped potential in medicinal and agrochemical applications. However, for most of these discovered compounds, the lack of deep pharmacological mechanisms and comprehensive pharmacokinetic evaluation limits their applications. Overall, this review shed light on the new secondary metabolites from the *Xylaria* fungi for their potential contributions to the future development of new natural product drugs in the agricultural and medicinal fields.

## Figures and Tables

**Figure 1 jof-10-00190-f001:**
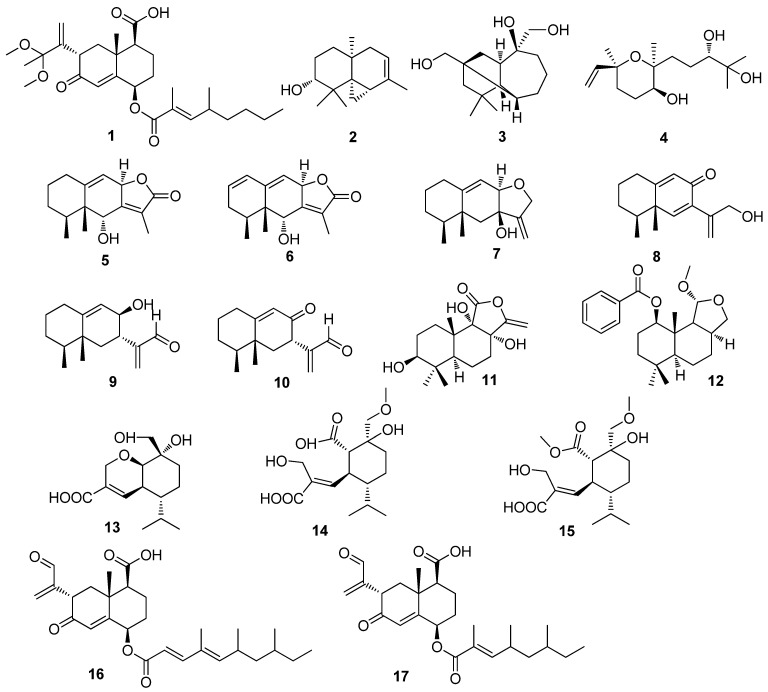
Chemical structures of sesquiterpenes **1**–**84** from *Xylaria* spp.

**Figure 2 jof-10-00190-f002:**
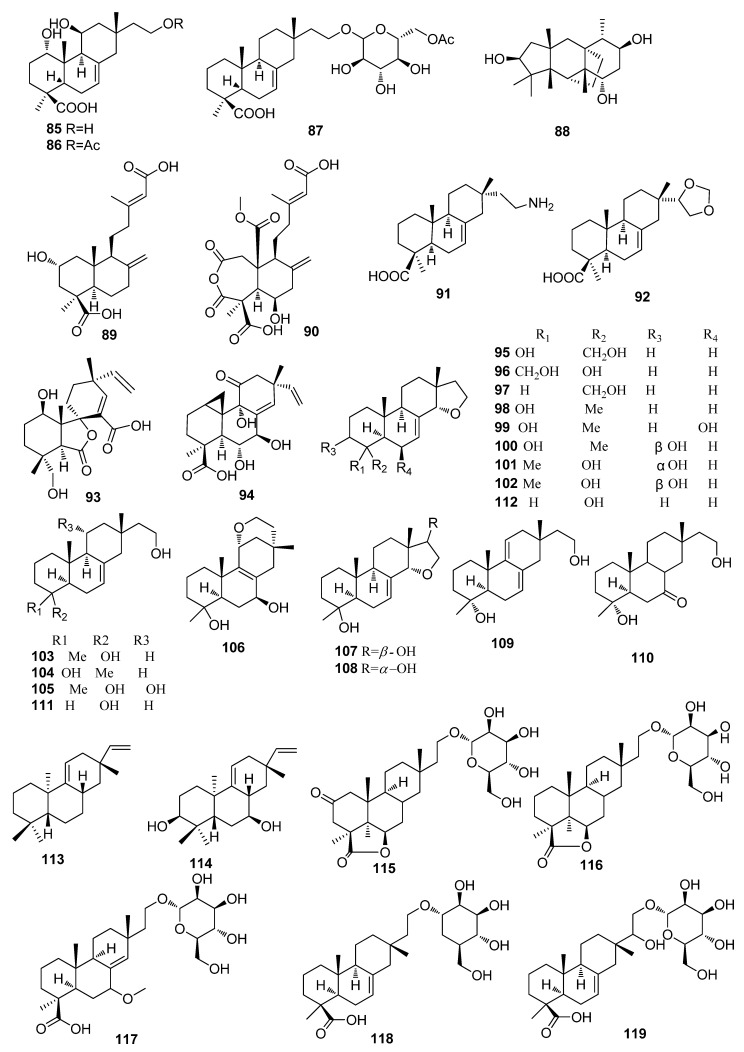
Chemical structures of diterpenes **85**–**127** from *Xylaria* spp.

**Figure 3 jof-10-00190-f003:**
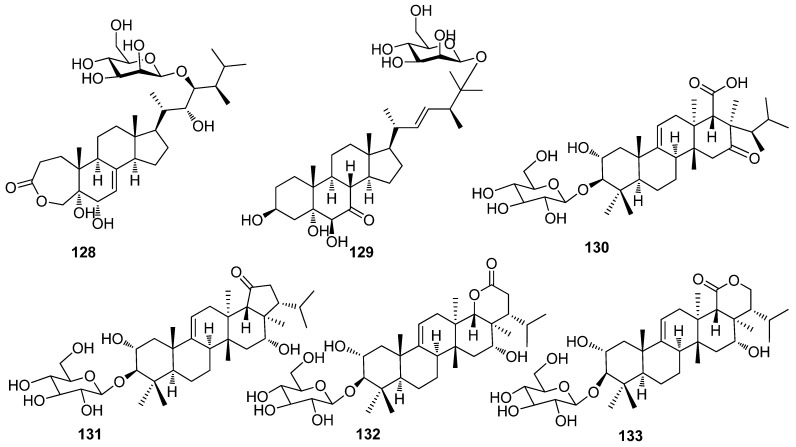
Chemical structures of triterpenoids **128**–**133** from *Xylaria* spp.

**Figure 4 jof-10-00190-f004:**
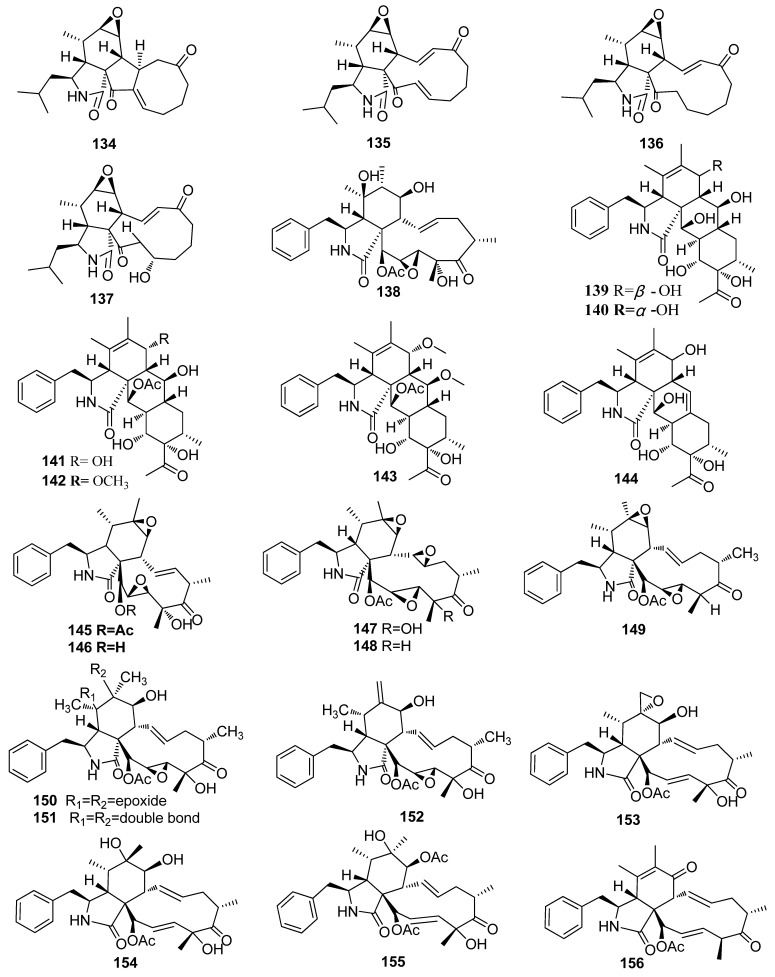
Chemical structures of cytochalasans **134**–**200** from *Xylaria* spp.

**Figure 5 jof-10-00190-f005:**
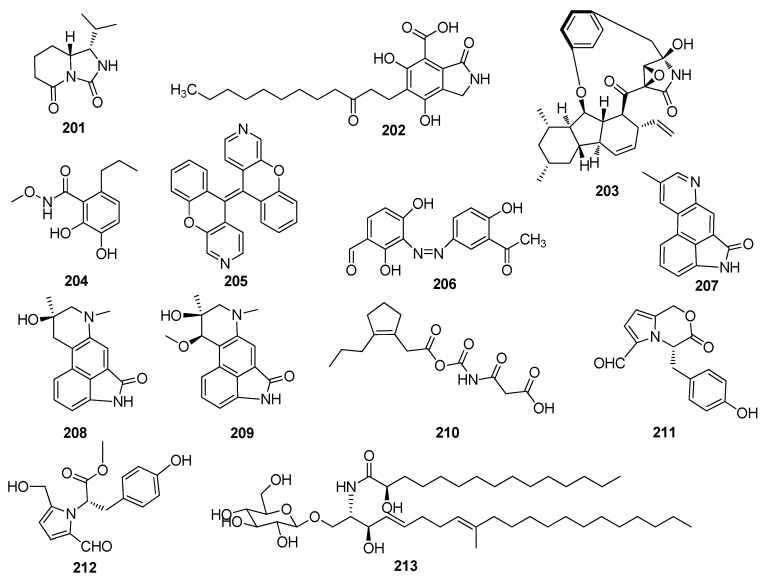
Chemical structures of other nitrogen-containing metabolites **201**–**245** from *Xylaria* spp.

**Figure 6 jof-10-00190-f006:**
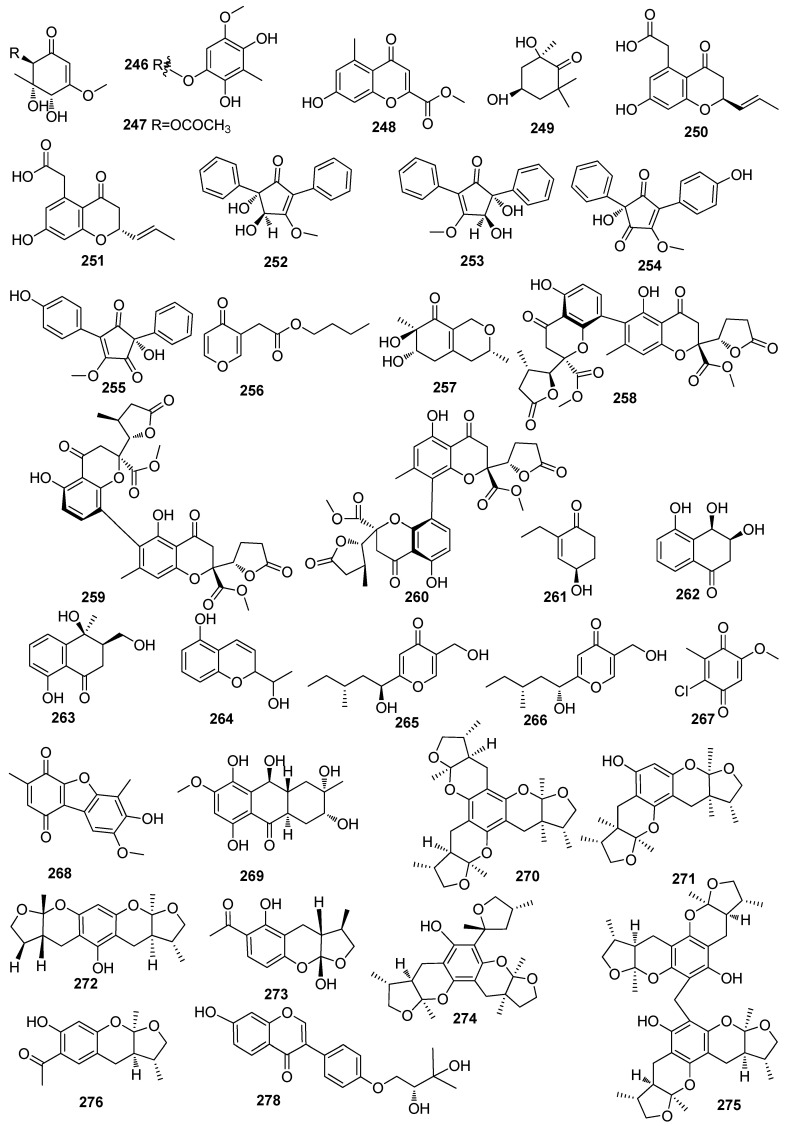
Chemical structures of polyketides **246**–**315** from *Xylaria* spp.

**Figure 7 jof-10-00190-f007:**
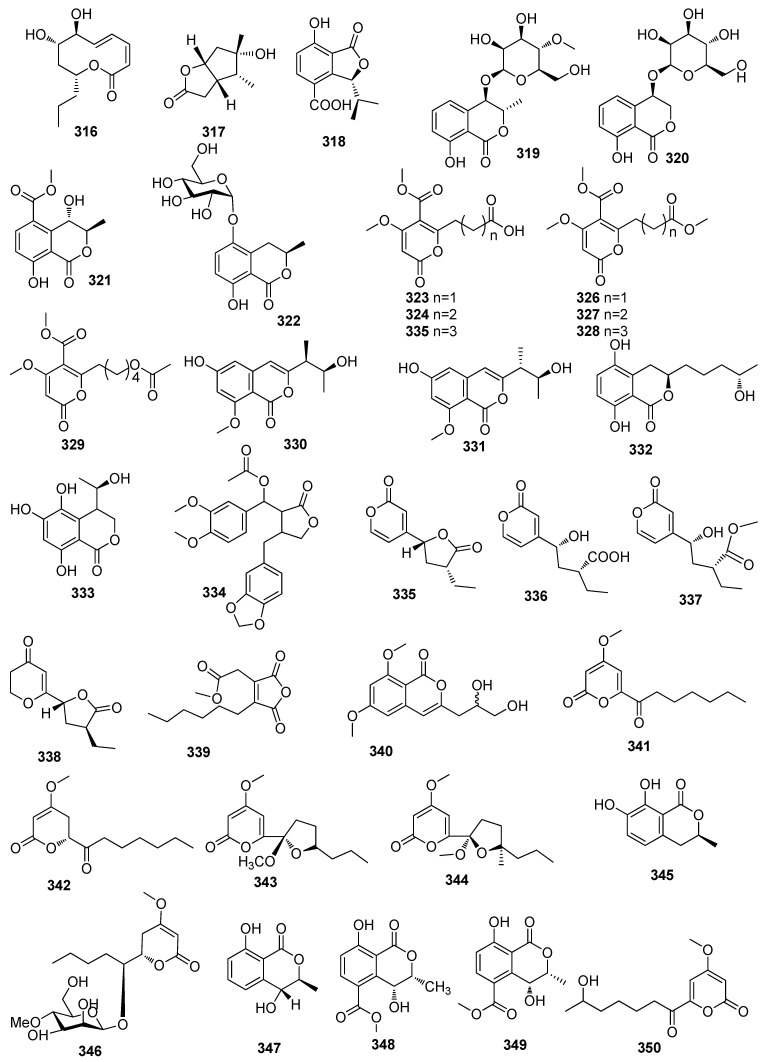
Chemical structures of lactones **316**–**390** from *Xylaria* spp.

**Figure 8 jof-10-00190-f008:**
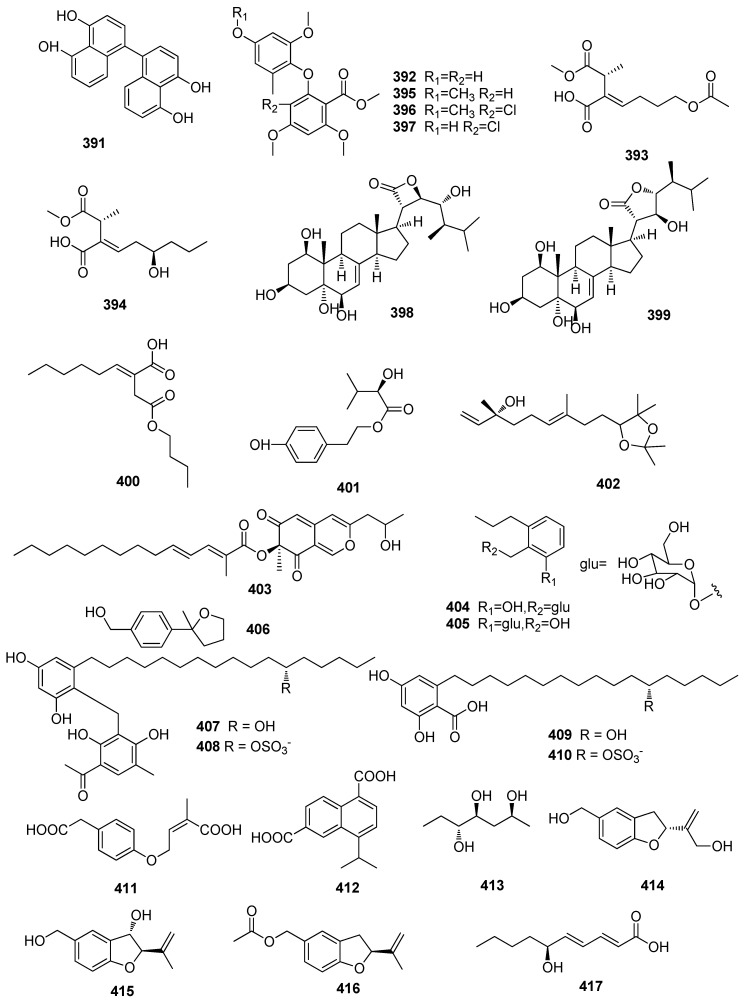
Chemical structures of polyketides **391**–**445** from *Xylaria* spp.

**Figure 9 jof-10-00190-f009:**
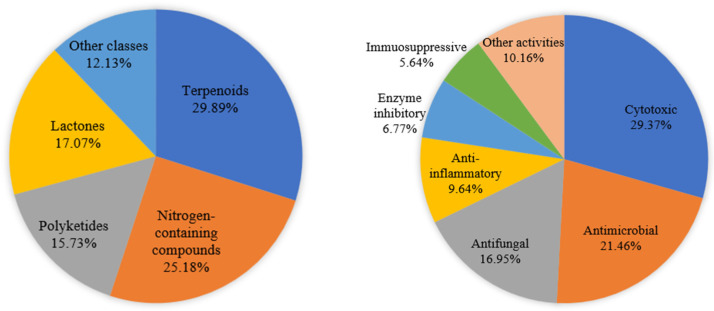
Structural diversity (**left**) and bioactivities (**right**) of secondary metabolites in the genus of *Xylaria* that were discovered from 1994 to January 2023.

**Table 1 jof-10-00190-t001:** The bioactivities of secondary metabolites **1**–**445** from *Xylaria* spp.

Compounds	Producing Strains	Habitats	Bioactivities	Refs
13,13-dimethoxyintegric acid (**1**)	*Xylaria* sp. V-27	Dead branch	Cytotoxicity	[10]
10-hydroxythujopsene (**2**)	*Xylaria cubensis* BCRC 09F 0035	*Litsea akoensis*	-	[11]
Akotriol (**3**)	*Xylaria cubensis* BCRC 09F 0035	*Litsea akoensis*	-	[11]
Xylaritriol (**4**)	*Xylaria cubensis* BCRC 09F 0035	*Litsea akoensis*	-	[11]
Nigriterpenes A, B (**5**–**6**), and D–F (**8**–**10**)	*X. nigripes* YMJ653	*Termite nest*	-	[12]
Nigriterpene C (**7**)	*X. nigripes* YMJ653	*Termite nest*	Anti-inflammatory activity	[12]
Polymorphine A (**11**)	*Xylaria polymorpha* (Pers.: Fr.)	Unknown	-	[13]
Polymorphine B (12)	*Xylaria polymorpha* (Pers.: Fr.)	Unknown	AChE inhibitory activity	[13]
Xylaric acids A–C (**13**–**15**)	*Xylaria* sp.	*Termite nest*	-	[14]
Eremoxylarins A (**16**) and B (**17**)	*Xylaria* sp. (YUA-026)	Unidentified plant	Antibacterial activity	[15]
Eremoxylarin C (**18**)	*Xylaria allantoidea* BCC 23163	*Decaying wood*	Cytotoxicity	[16]
Eremoxylarins D (**19**), F (**20**), and G (**21**)	*Xylaria allantoidea* BCC 23163	*Rhizocarpon geographicum*	Antibacterial activity	[17]
Eremoxylarin E (**22**) and H (**23**)	*Xylaria allantoidea* BCC 23163	*Rhizocarpon geographicum*	-	[17]
Eremoxylarin I (**24**)	*Xylaria allantoidea* BCC 23163	*Rhizocarpon geographicum*	Antibacterial activity, cytotoxicity	[17]
Eremoxylarin J (**25**)	*Xylaria allantoidea* BCC 23163	*Rhizocarpon geographicum*	-	[17]
10*α*-Hydroxyeremophil-7(11)-en-2,3:12,8-diolide (**26**)	*Xylaria* sp. BCC 60404	Mangrove plant	-	[18]
1*β*-Acetoxy-10*α*,13-dihydroxyeremophil-7(11)-en-12,8*β*-olide (**27**)	*Xylaria* sp. BCC 60404	Mangrove plant	-	[18]
1*α*,10*α*-Epoxy-2*α*,13-dihydroxyeremophil-7(11)-en-12,8*β*-olide (**28**)	*Xylaria* sp. BCC 60404	Mangrove plant	-	[18]
1*α*,10*α*-Epoxy-2*β*,13-dihydroxyeremophil-7(11)-en-12,8*β*-olide (**29**)	*Xylaria* sp. BCC 60404	Mangrove plant	-	[18]
1*α*,10*α*-Epoxy-3*α*,13-dihydroxyeremophil-7(11)-en-12,8*β*-olide (**30**)	*Xylaria* sp. BCC 60404	Mangrove plant	-	[18]
1*α*,10*α*-Epoxy-3*β*,13-dihydroxyeremophil-7(11)-en-12,8*β*-olide (**31**)	*Xylaria* sp. BCC 60404	Mangrove plant	Cytotoxicity	[18]
1*α*,10*α*,2*α*,3*α*-Diepoxyeremophil-7(11)-en-12,8β-olide (**32**)	*Xylaria* sp. BCC 60404	Mangrove plant	-	[18]
2-Oxo-13-hydroxyeremophila-1(10),7(11)-dien-12,8*β*-olide (13- hydroxyxylareremophil (**33**)	*Xylaria* sp. BCC 60404	Mangrove plant	-	[18]
7-Epi-tessaric acid (**34**)	*Xylaria* sp. BCC 60404	Mangrove plant	-	[18]
2*β*-Hydroxyeremophila-1(10),11(13)-dien-12-oic acid (**35**)	*Xylaria* sp. BCC 60404	Mangrove plant	-	[18]
Xylcarpins A–E (**36**–**40**)	*Xylaria carpophila* (Pers.)	Unknown	-	[19]
Xylarioxides A–C (**41**–**43**)	*Xylaria* sp. YM 311647	*Azadirachta indica*	Antifungal activity	[20]
Xylarioxide D (**44**)	*Xylaria* sp. YM 311648	*Azadirachta indica*	-	[20]
Xylareremophil (**45**)	*Xylaria* sp. (GDG-102)	Mangrove plant	Antibacterial activity	[21]
Xylarenones A (**46**) and B (**47**)	*Xylaria* sp. (NCY2)	Unknown	Cytotoxicity	[22]
Xylarenic acid (**48**)	*Xylaria* sp. (NCY2)	Unknown	Cytotoxicity	[22]
(1*S*,2*S*,4*S*,5*S*,7*R*,10*R*)-Guaiane-2,10,11,12-tetraol (**49**)	*Xylaria* sp. (YM311647)	*Azadirachta indica*	Antibacterial activity	[23]
(1*S*,2*S*,4*S*,5*S*,7*R*,10*R*)-Guaiane-2,4,10,11,12-pentaol (**50**)	*Xylaria* sp. (YM311647)	*Azadirachta indica*	Antibacterial activity	[23]
(1*S*,4*R*,5*S*,7*R*,10*R*)-Guaiane-4,5,10,11,12-pentaol (**51**)	*Xylaria* sp. (YM311647)	*Azadirachta indica*	Antibacterial, antifungal activity	[23]
(1*R*,4*S*,5*R*,7*R*,10*R*)-Guaiane-1,5,10,11,12-pentaol (**52**)	*Xylaria* sp. (YM311647)	*Azadirachta indica*	Antibacterial, antifungal activity	[23]
(1*R*,4*R*,5*R*,7*R*,10*R*)-11-Methoxyguaiane-4,10,12-triol (**53**)	*Xylaria* sp. (YM311647)	*Azadirachta indica*	Antibacterial activity	[23]
1*β*,7*α*,10*α*-Trihydroxyeremophil-11(13)-en-12,8*β*-olide (**54**)	*Xylaria* sp. (BCC 21097)	Mangrove plant	Cytotoxicity	[24]
7*α*,10*α*-Dihydroxy-1*β*-methoxyeremophil-11(13)-en-12,8*β*-olide) (**55**)	*Xylaria* sp. (BCC 21097)	Mangrove plant	Cytotoxicity, antimalarial activity	[24]
1α,10*α*-Epoxy-7α-hydroxyeremophil-11(13)-en-12,8*β*-olide (**56**)	*Xylaria* sp. (BCC 21097)	Mangrove plant	Cytotoxicity, antimalarial, antifungal activity	[24]
1*β*,10α,13-Trihydroxyeremophil-7(11)-en-12,8-olide (**57**)	*Xylaria* sp. (BCC 21097)	Mangrove plant	-	[24]
10*β*,13-Dihydroxy-1-methoxyeremophil-7(11)-en-12,8*β*-olide (**58**)	*Xylaria* sp. (BCC 21097)	Mangrove plant	-	[24]
Mairetolide F (**59**)	*Xylaria* sp. (BCC 21097)	Mangrove plant	-	[24]
1*β*,10*α*-Epoxy-13-hydroxyeremophil-7(11)-en-12,8*β*-olide (**60**)	*Xylaria* sp. (BCC 21103)	Mangrove plant	-	[24]
1*β*,10*α*-Epoxy-3r-hydroxyeremophil-7(11)-en-12,8*β*-olide (**61**)	*Xylaria* sp. (BCC 21104)	Mangrove plant	-	[24]
12,8-Eudesmanolides 3*α*,4*α*,7*β*-trihydroxy-11(13)-eudesmen-12,8-olide (**62**)	*Xylaria ianthinovelutina*	*Torreya jackii* Chun	-	[25]
4*α*,7*β*-Dihydroxy-3*α*-methoxy-11(13)-eudesmen-12,8-olide (**63**)	*Xylaria ianthinovelutina*	*Torreya jackii* Chun	-	[25]
7*β*-Hydroxy-3,11(13)-eudesmadien-12,8-olide (**64**)	*Xylaria ianthinovelutina*	*Torreya jackii* Chun	-	[25]
13-Hydroxy- 3,7(11)-eudesmadien-12,8-olide (**65**)	*Xylaria ianthinovelutina*	*Torreya jackii* Chun	-	[25]
9,15-Dihydroxy-presilphiperfolan-4-oic acid (**66**)	*Xylaria* sp. YM 311647	*Azadirachta indica*	-	[26]
15-Acetoxy-9-hydroxy-presilphiperfolan-4-oic acid (**67**)	*Xylaria* sp. YM 311647	*Azadirachta indica*	-	[26]
Eremophilane sesquiterpenes (**68**–**70**)	*Xylaria* sp. BL321.	*Licuala spinosa*	-	[27]
Xylaguaianols A−D (**71**–**74**)	*Xylaria* sp. NC1214	Unidentified seed	-	[28]
Isocadinanol A (**75**)	*Xylaria* sp. NC1214	Unidentified seed	-	[28]
(1*S*,4*S*,5*R*,7*R*,10*R*,11*R*)-Guaiane-5,10,11,12-tetraol (**76**)	*Xylaria* sp. YM 311647	leaves of *Piper aduncum*	Antibacterial activity	[29]
(1*S*,4*S*,5*R*,7*R*,10*R*,11*S*)-Fuaiane-1,10,11,12-tetraol (**77**)	*Xylaria* sp. YM 311647	leaves of *Piper aduncum*	Antibacterial activity	[29]
(1*S*,4*S*,5*R*,7*R*,10*R*,11*S*)-guaiane-5,10,11,12-tetraol (**78**)	*Xylaria* sp. YM 311647	leaves of *Piper aduncum*	Antibacterial activity	[29]
(1*S*,4*S*,5*S*,7*R*,10*R*,11*R*)-guaiane-1,10,11,12-tetraol (**79**)	*Xylaria* sp. YM 311647	leaves of *Piper aduncum*	Antibacterial activity	[29]
(1*R*,3*S*,4R,5S,7R,10R,11*S*)-Guaiane-3,10,11,12-tetraol (**80**)	*Xylaria* sp. YM 311647	leaves of *Piper aduncum*	Antibacterial activity	[29]
(1*R*,3*R*,4*R*,5*S*,7*R*,10*R*,11*R*)-Guaiane-3,10,11,12-tetraol (**81**)	*Xylaria* sp. YM 311647	leaves of *Piper aduncum*	Antibacterial activity	[29]
(1*R*,4*S*,5*S*,7*S*,9*R*,10*S*,11*R*)-Guaiane-9,10,11,12-tetraol (**82**)	*Xylaria* sp. YM 311647	leaves of *Piper aduncum*	Antibacterial activity	[29]
(1*R*,4*S*,5*S*,7*R*,10*R*,11*S*)-Guaiane-10,11,12-triol (**83**)	*Xylaria* sp. YM 311647	leaves of *Piper aduncum*	Antibacterial activity	[29]
(1*R*,4S,5S,7*R*,10*R*,11*R*)-Guaiane-10,11,12-triol (**84**)	*Xylaria* sp. YM 311647	leaves of *Piper aduncum*	Antibacterial activity	[29]
Xylongoic acids A–C (**85**–**87**)	*Xylaria longipes* HFG1018	*Fomitopsis betulina*	-	[30]
Diterpenoid cubentriol (**88**)	*Xylaria cubensis* BCRC 09F 0035	*Litsea akoensis*	-	[11]
Hypoxyterpoids A (**89**)	*Xylaria cubensis*	*Bruguiera gymnorrhiza*	-	[31]
Hypoxyterpoids B (**90**)	*Xylaria cubensis*	*Bruguiera gymnorrhiza*	*α*-glucosidase inhibitory activity	[31]
Xylarianes A (**91**) and B (**92**)	*Xylaria* sp. 290	Unkonwn	-	[32]
Spiropolin A (**93**)	*Xylaria polymorpha*	Wild mushroom	-	[33]
Myrocin E (**94**)	*Xylaria polymorpha*	Wild mushroom	-	[33]
Xylarinorditerpenes A (**95**), F–H (**100**–**102**), J–M (**104**–**107**), and O–R (**109**–**112**)	*Xylaria longipes* HFG1018	*Fomitopsis betulinus*	-	[34]
Xylarinorditerpenes B–E (**96**–**99**), I (103), and N (**108**)	*Xylaria longipes* HFG1018	*Fomitopsis betulinus*	Immunosuppressive activity	[34]
Acanthoic acid (**113**)	*Xylaria* sp. (EJCP07)	Unkonwn	Antibacterial activity	[35]
3*β*,7*β*-Dihydroxyacanthoic acid (**114**)	*Xylaria* sp. (EJCP07)	Unkonwn	Antibacterial activity	[35]
Xylarcurcosides A–C (**115**–**117**)	*Xylaria curta* YSJ-5	Leaves of *Alpinia zerumbet*	-	[36]
16-*α*-*D*-mannopyranosyloxyisopimar-7-en-19-oic acid (**118**)	*Xylaria polymorpha*	Fruit bodies	Cytotoxicity activity	[37]
15-Hydroxy-16-*α*-*D*-mannopyranosyloxyisopimar-7-en-19-oic acid (**119**)	*Xylaria polymorpha*	Fruit bodies	Cytotoxicity activity	[37]
16-*α*-*D*-glucopyranosyloxyisopimar-7-en-19-oic acid (**120**)	*Xylaria polymorpha*	Fruit bodies	Cytotoxicity activity	[37]
Xylabisboeins A (**121**) and B (**122**)	*Xylaria* sp. SNB-GTC2501	Unknown	-	[38]
14*α*,16-epoxy-18-norisopimar-7-en-4*α*-ol (**123**)	*Xylaria* sp. YM 311647	*Licuala spinosa*	-	[27]
16-*O*-Sulfo-18-norisopimar-7-en-4a,16-diol (**124**)	*Xylaria* sp. YM 311647	*Licuala spinosa*	Antifungal activity	[27]
9-Deoxy-hymatoxin A (**125**)	*Xylaria* sp. YM 311647	*Licuala spinosa*	Antibacterial activity	[27]
Xylarilongipin A (**126**)	*Xylaria longipes* HFG1018	Leaves of *Alpinia zerumbet*	Cytotoxicity	[39]
Xylarilongipin B (**127**)	*Xylaria longipes* HFG1018	Leaves of *Alpinia zerumbet*	-	[39]
Xylarioxides E–F (**128**–**129**)	*Xylaria* sp. YM 311647	*A. indica*	Antibacterial activity	[20]
Kolokoside A (**130**)	*Xylaria* sp.	Fruit bodies	Antibacterial activity	[40]
Kolokosides B–D (**131**–**133**)	*Xylaria* sp.	Fruit bodies	-	[40]
Lagambasines A–D (**134**–**137**)	*Xylaria* sp. WH2D4	Fruit bodies	-	[41]
Karyochalasin A (**138**)	*X. karyophthora*	Unknown	-	[42]
Curtachalasins X1 (**139**), X5 (**143**)	*Xylaria curta* E10	*Solanum tuberosum*	Cytotoxicity	[43]
Curtachalasins X2-X4 (**140**–**142**)**,** X6 (**144**)	*Xylaria curta* E10	*Solanum tuberosum*	-	[43]
19,20-Epoxycytochalasin Q (**145**)	*Xylaria obovate*	*Decaying wood*	Cytotoxicity	[44]
Deacetyl-19,20-epoxycytochalasin Q (**146**)	*Xylaria obovate*	*Decaying wood*	-	[44]
Eytoehalasins 19,20-epoxycytochalasin R (**147**)	*Xylaria hypoxylon*	Unknown	-	[45]
18-Deoxy-19,20-epoxycytochalasin R (**148**)	*Xylaria hypoxylon*	Unknown	-	[45]
18-Deoxy-19,20-epoxycytochalasin Q (**149**)	*Xylaria hypoxylon*	Unknown	-	[45]
19,20-Epoxycytochalasin N (**150**)	*Xylaria hypoxylon*	Unknown	-	[45]
19,20-Epoxycytochalasin C (**151**)	*Xylaria hypoxylon*	Unknown	-	[45]
21-Acetylengleromycin (**152**)	*Xylaria hypoxylon*	Unknown	-	[45]
6,12-Epoxycytochalasin D (**153**)	*Xylaria longipes*	Fruit bodies	-	[46]
6-Epi-cytochalasin P (**154**)	*Xylaria longipes*	Fruit bodies	-	[46]
7-*O*-acetylcytochalasin P (**155**)	*Xylaria longipes*	Fruit bodies	-	[46]
7-Oxo-cytochalasin C (**156**)	*Xylaria longipes*	Fruit bodies	-	[46]
12-Hydroxylcytochalasin Q (**157**)	*Xylaria longipes*	Fruit bodies	-	[46]
Curtachalasin Q (**158**)	*Xylaria* sp. DO1801	*Solanum tuberosum*	-	[47]
19-Epi-cytochalasin P1 (**159**)	*Xylaria* cf. *Curta*	Soil	Cytotoxicity	[48]
6-Epi-19,20-epoxycytochalasin P (**160**)	*Xylaria* cf. *Curta*	Soil	-	[48]
7-*O*-acetyl-6-epi-19,20-epoxycytochalasin P (**161**)	*Xylaria* cf. *Curta*	Soil	Cytotoxicity	[48]
7-*O*-acetyl-19-epi-cytochalasin P1 (**162**)	*Xylaria* cf. *Curta*	Soil	Cytotoxicity	[48]
6-*O*-acetyl-6-epi-19,20-epoxycytochalasin P (**163**)	*Xylaria* cf. *Curta*	Soil	-	[48]
7-*O*-acetyl-19,20-epoxycytochalasin C (**164**)	*Xylaria* cf. *Curta*	Soil	-	[48]
7-*O*-acetyl-19,20-epoxycytochalasin D (**165**)	*Xylaria* cf. *Curta*	Soil	Cytotoxicity	[48]
Deacetyl-5,6-dihydro-7-oxo-19,20-epoxycytochalasin C (**166**)	*Xylaria* cf. *Curta*	Soil	-	[48]
18-Deoxy-21-oxo-deacetyl-19,20-Epoxycytochalasin N (**167**)	*Xylaria* cf. *Curta*	Soil	-	[48]
Arbuschalasins A–D (**168**–**171**)	*Xylaria arbuscula* GZS74	*Bruguiera gymnorrhiza*	-	[49]
Xylarchalasin A (**172**)	*Xylaria* sp. GDGJ-77B	*Sophora tonkinensis*	-	[50]
Xylarchalasin B (1**73**)	*Xylaria* sp. GDGJ-77B	*Sophora tonkinensis*	Antibacterial activity	[50]
Curtachalasins A (**174**) and B (**175**)	*Xylaria curta* (E10)	*Solanum tuberosum*	Antibacterial activity	[51]
Cytochalasin P1(**176**)	*Xylaria* sp. SOF11	*Marine sediment*	Cytotoxicity	[52]
18-Deoxycytochalasin Q (**177**)	*Xylaria* sp. SCSIO156	*Marine sediment*	-	[53]
21-*O*-deacetylcytochalasin Q (**178**)	*Xylaria* sp. SCSIO156	*Marine sediment*	Cytotoxicity	[53]
Xylastriasan A (**179**)	*Xylaria striata*	Unknown	Cytotoxicity	[54]
Cytochalasin H2 (**180**)	*Xylaria* sp. (A23)	*Annona squamosa*	Cytotoxicity	[55]
Xylarichalasin A (**181**)	*Xylaria* cf. *curta*	*Solanum tuberosum*	Cytotoxicity	[56]
Cytochalasins D1 (**182**) and C1 (**183**)	*Xylaria* cf. *curta*	Unknown	Cytotoxicity	[57]
Cytochalasans (**184**–**188**)	*Xylaria longipes*	Unknown	-	[46]
Curtachalasin F(**189**)	*Xylaria* cf. *curta*	Unknown	Cytotoxicity	[58]
Curtachalasins G–N (**190**–**197**)	*Xylaria* cf. *curta*	Unknown	-	[58]
Curtachalasin O (**198**)	*Xylaria* cf. *curta*	Unknown	Cytotoxicity	[58]
Curtachalasin P (**199**)	*Xylaria* cf. *curta*	Unknown	-	[58]
Xylarisin B (**200**)	*Xylaria* sp. HNWSW-2	*Xylocarpus granatum*	-	[59]
Akodionine (**201**)	*Xylaria cubensis*	*Litsea akoensis*	-	[11]
Xylactam B (**202**)	*Xylaria* sp.	Leaves of *Tectaria zeylanica*	-	[60]
Xylarialoid A (**203**)	*Xylaria arbuscula*	Leaves of the plant *Rauvolfia vomitoria*	-	[61]
2,3-Dihydroxy-N-methoxy-6-propylbenzamide (**204**)	*Xylaria* sp. PSU-H182	*Hevea brasiliensis*	-	[62]
Xylopyridine A (**205**)	*Xylaria* sp.	Unidentified plant	Cytotoxicity	[63]
(*Z*)-3-{(3-acetyl-2-hydroxyphenyl) diazenyl}-2,4-dihydroxybenzaldehyde (**206**)	*Xylaria psidii*	*Amandinea medusulina*	Cytotoxicity	[64]
Xylanigripones A (**207**)	*Xylaria nigripes* (KL.)	Unidentified plant	Inhibition of CEPT activity	[65]
Xylanigripones B-C (**208**–**209**)	*Xylaria nigripes* (KL.)	Unidentified plant	-	[65]
Xylariahgin F (**210**)	*Xylaria* sp.	*Isodon sculponeatus*	-	[66]
(4*S*)-3,4-dihydro-4-(4-hydroxybenzyl)-3-oxo-1H-pyrrolo [2,1-*c*][1,4]oxazine-6-carbaldehyde (**211**)	*Xylaria nigripes*	*Termite nest*	-	[67]
Methyl (2*S*)-2-[2-formyl-5-(hydroxymethyl)-1H-pyrrol-1-yl]-3-(4-hydr-oxyphenyl)propanate (**212**)	*Xylaria nigripes*	*Termite nest*	-	[67]
Allantoside (**213**)	*Xylaria allantoidea* SWUF76	Unknown	-	[68]
Sinuxylamides A–B (**214**–**215**)	*Xylaria* sp. FM1005	*Sinularia densa*	Cytotoxicity	[69]
Sinuxylamides C–D (**216**–**217**)	*Xylaria* sp. FM1005	*Sinularia densa*	-	[69]
Sssinuxylamide E (**218**)	*Xylaria* sp. FM1005	*Sinularia densa*	-	[69]
4-(7,8-Dihydroxy-4-oxoquinazolin-3(4H)-yl)butanoic acid (**219**)	*Xylaria* sp. FM1005	*Sinularia densa*	-	[69]
4-(8-Hydroxy-4-oxoquinazolin-3(4H)-yl)butanoic acid (**220**	*Xylaria* sp. FM1005	*Sinularia densa*	-	[69]
3,4-Dihydroisocoumarin derivative 1′-N-Acetyl-5-methylmellein (**221**)	*Xylaria* sp. FM1005	*Sinularia densa*	-	[69]
Xylariamide (**222**)	*Xylaria plebeja* PSU-G30	*Garcinia hombroniana*	-	[70]
Xylaramide (**223**)	*Xylaria longipes*	Wood	Antifungal activity	[71]
2,5-Diamino-N-(1-amino-1-imino-3-methylbutan-2-yl) pentanamide (**224**)	*Xylaria* cf. *cubensis* SWUF08-86	*Decaying wood*	-	[72]
Xylariamino acid A (**225**)	*Xylaria nigripes* (Kl.)	Unknown	-	[73]
Xylapyrroside A (**226**)	*Xylaria nigripes*	Wuling powder	Antibacterial activity	[74]
Xylapyrroside B (**227**)	*Xylaria nigripes*	Wuling powder	-	[74]
(±)-Xylaridines A and B (**228**–**229**)	*Xylaria longipes*	Unknown	Antibacterial activity	[75]
(−)-Xylariamide A (**230**)	*Xylaria* sp.	*Glochidion ferdinandi*	Insect resistance activity	[76]
Cyclotripeptide X-13 (**231**)	*Xylaria* sp. (No. 2508)	Mangrove	Angiogenic property	[77,78]
Xyloallenoide A (**232**)	*Xylaria* sp. (No. 2508)	Mangrove	Angiogenic property	[77,78]
Xyloallenoide A1 (**233**)	*Xylaria* sp. (No. 2508)	Mangrove	Angiogenic property	[77,78]
Cyclotripeptide X-13a (**234**	*Xylaria* sp. (No. 2508)	Mangrove	Angiogenic property	[77,78]
Xylaroamide A (**235**)	*Xylaria* sp. 218-066	Mangrove	Cytotoxicity	[79]
Xylarotides A (**236**) and B (**237**)	*Xylaria* sp. 101.	Gaoligong Mountain	-	[80]
Xylapeptide A (**238**)	*Xylaria* sp. GDG-102	*Sophora tonkinensisan*	Antibacterial activity	[81]
Xylapeptide B (**239**)	*Xylaria* sp. GDG-102	*Sophora tonkinensisan*	Antibacterial activity	[81]
Ellisiiamide A **240**	*Xylaria ellisii*	Blueberry *Vaccinium angustifolium*	Antibacterial activity	[82]
Ellisiiamides B–C (**241**–**242**)	*Xylaria ellisii*	Blueberry *Vaccinium angustifolium*	-	[82]
Cyclo(*N*-methyl-*L*-Phe-*L*-Val-*D*-Ile-*L*-Leu*-L*-Pro (**243**)	*Xylaria* sp.	Lichen *Leptogium saturninum*	Antifungal activity	[83]
Cyclo(*L*-Val-*D*-Ile-*L*-Leu-*L*-pro-D-Leu (**244**)	*Xylaria* sp.	Lichen *Leptogium saturninum*	Antifungal activity	[83]
Pentaminolarin (**245**)	*Xylaria* sp. (SWUF08-37)	Wood-decaying	Cytotoxicity	[84]
Xylariacyclones A (**246**) and B (**247**)	*Xylaria plebeja* PSU-G30	*Garcinia hombroniana*	-	[74]
Xylarianin B (**248**)	*Xylaria* sp. SYPF 8246.	*Panax notoginseng*	-	[85]
Xylariaone (**249**)	*Xylaria* sp. 12F075	Lichen *Leptogium saturninum*	-	[86]
(+)-Xylarichromone A (**250**)	*Xylaria nigripes*	*Decaying wood*	Cytotoxicity	[87]
(-)-Xylarichromone A (**251**)	*Xylaria nigripes*	*Decaying wood*	-	[87]
Xylariaones A1-B2 (**252**–**255**)	*Xylaria* sp.	*Cudrania tricuspidata*	-	[88]
Xylaripyone H (**256**)	*Xylaria* sp.	*Cudrania tricuspidata*	-	[88]
Xylariphilone (**257**)	*Xylaria* sp. PSU-ES163	Seagrass *Halophila ovalis*	-	[89]
Xylaromanones A–C (**258**–**260**)	*Xylaria* sp. PSU-H182	*Hevea brasiliensis*	-	[62]
(*R*)-4-Hydroxy-2-ethyl-2-cyclohexen-1-one (**261**)	*Xylaria* sp. PSU-H182	*Hevea brasiliensis*	-	[63]
3,4,5-Trihydroxy-1-tetralone (**262**)	*Xylaria* sp.	*Termite nest*-derived	-	[14]
Hemi-cycline A (**263**)	*Xylaria* cf. *cubensis* SWUF08-86	Unknown	-	[76]
Hexacycloxylariolone (**264**)	*Xylaria* sp.	Unknown	Cytotoxicity	[90]
Xylaropyrones B (**265**) and C (**266**)	*Xylaria* sp. SC1440	*Spartina maritima*	-	[91]
2-Chloro-5-methoxy-3-methylcyclohexa-2,5-diene-1,4-dione (**267**)	*Xylaria* sp.	*Sandoricum koetjape*	Antifungal activity, cytotoxicity	[92]
Xylariaquinone A (**268**)	*Xylaria* sp.	*Sandoricum koetjape*	Antifungal activity	[92]
Xylanthraquinone (**269**)	*Xylaria* sp. (No. 2508)	Mangrove	-	[93,94,95,96,97,98]
Xyloketal A (**270**)	*Xylaria* sp. (No. 2508)	Mangrove	Anti-inflammatory	[93,94,95,96,97,98]
Xyloketal B(**271**)	*Xylaria* sp. (No. 2508)	Mangrove	Anti-inflammatory, cytotoxicity, NO disturbance, intracellular Ca^2+^ imbalance	[93,94,95,96,97,98]
Xyloketals C–H (**272**–**277**)	*Xylaria* sp. (No. 2508)	Mangrove	Anti-inflammatory	[93,94,95,96,97,98]
Xyloketal J (**278**)	*Xylaria* sp. (No. 2508)	Mangrove	Anti-inflammatory	[93,94,95,96,97,98]
Paecilins F–K (**279**–**284**, **286**)	*Xylaria curta* E10	Potato tissues	-	[99]
Paecilins L and N (**285** and **287**)	*Xylaria curta* E10	Potato tissues	Antibacterial	[99]
Paecilins O–P (**288**–**289**)	*Xylaria curta* E10	Potato tissues	-	[99]
Rubiginosins A–C (**290**–**292**)	*Xylariaceus ascomycete*	*Fraxinus excelsior*	-	[100]
Xylaphenoside A (**293**)	*Xylaria* sp. CGMCC No. 5410	*Selaginella moellendorffii*	Antimicrobial	[101]
Xylarinonericins A–C (**294**–**296**)	*Xylaria plebeja* PSU-G30	Wood	-	[71]
Rubiginosins A–C (**297**–**299**)	*Xylariaceus ascomycete*	Fruit bodies	-	[102]
1,3,8-Trihydroxy-7-methoxy-9-methyldibenzofuran (**300**)	*Xylaria feejeensis*	*Geophila repens*	Cytotoxicity	[103]
(6*S*,2′*R*,6′*S*)-6-Methyl-2-((6-methyltetrahydro-2H-pyran-2-yl)methyl)-2,3-dihydro-4H-pyran-4-one (**301**)	*Xylaria feejeensis*	*Geophila repens*	-	[103]
(2′*R*,6′*S*)-5-((-6-Methyltetrahydro-2H-pyran-2-yl)methyl)benzene-1,3-diol (**302**)	*Xylaria feejeensis*	*Geophila repens*	Cytotoxicity	[103]
6′,7′-Didehydrointegric acid (**303**)	*Xylaria feejeensis*	*Geophila repens*	-	[104]
13-Carboxyintegric acid (**304**)	*Xylaria feejeensis*	*Geophila repens*	-	[104]
(4*S*,5*S*,6*S*)-5,6-epoxy-4-hydroxy-3-methoxy-5-methyl-cyclohex-2-en-1-one (**305**)	*Xylaria carpophila*	*Ligustrum lucidum*	Cytotoxicity	[19]
Xylariols A (**306**) and B (**307**)	*Xylaria hypoxylon* AT-028	*Ligustrum lucidum*	Cytotoxicity	[105]
1-(Xylarenone A)xylariate A (**308**)	*Xylaria* sp. NCY2	*Torreya jacki*		[106]
Schweinitzin A (**309**)	*Xylaria schweinitzii* Berk. and M.A	*Geophila repens*	Cytotoxicity	[107]
Schweinitzin B (**310**)	*Xylaria schweinitzii* Berk. and M.A	*Geophila repens*	-	[107]
6-Ethyl-8-hydroxy-4H-chromen-4-one (**311**)	*Xylaria* sp. SWUF09-62	*Ligustrum lucidum*	-	[108]
6-Ethyl-7,8-dihydroxy-4H-chromen-4-one (**312**)	*Xylaria* sp. SWUF09-62	*Ligustrum lucidum*	Anti-inflammatory activity	[108]
Mellisol (**313**)	*Xylaria mellisii* (BCC 1005).	*Torreya jacki*	Antivirus activity	[109]
*γ*-pyrone-3-acetic acid (**314**)	*Xylatia*	Unknown	-	[110]
α-pyrone 9-hydroxyxylarone (**315**)	*Xylaria* sp. NC1214	Moss	-	[28]
Xylarolide (**316**)	*Xylaria* sp. 101	Mushroom	-	[80]
(3α*S*,6α*R*)-4,5-Dimethyl-3,3α,6,6v-Tetrahydro-2Hcyclopenta [β]furan-2-one (**317**)	*Xylaria curta* 92092022	Unknown	Antibacterial activity	[111]
Xylarphthalide A (**318**)	*Xylaria* sp. GDG-102	*Sophora tonkinensis*	Antibacterial activity	[112]
Xylarglycosides A (**319**) and B (**320**)	*Xylaria* sp. KYJ-15	*Illigera celebica*	Antibacterial, antioxidant activity	[113]
Akolitserin (**321**)	*Xylaria cubensis*	*Litsea akoensis*	-	[11]
5-*O*-*α*-Dglucopyranosyl-5-Hydroxymellein (**322**)	*Xylaria* sp. CGMCC No.5410	*Selaginella moellendorffii*	Antimicrobial activity	[101]
Xylaripyones A-C(**323**), E (**327**), G (**329**)	*Xylaria* sp.	*Cudrania tricuspidata*	-	[98]
Xylaripyone D (**326**)	*Xylaria* sp.	*Cudrania tricuspidata*	Cytotoxicity	[98]
Xylaripyone F (**328**)	*Xylaria* sp.	*Cudrania tricuspidata*	Anti-inflammatory activity	[98]
Hypoxymarins A, B, and D (**330**–**331** and **333**)	*Hypoxylon* sp. (Hsl2-6)	*Bruguiera gymnorrhiza*	-	[114]
Hypoxymarin C (**332**)	*Hypoxylon* sp. (Hsl2-6)	*Bruguiera gymnorrhiza*	Antioxidant activity	[114]
Xylariahgins A–E (**335**–**340**)	*Xylaria* sp. hg1009	*Isodon sculponeatus*		[66]
3-(2,3-Dihydroxypropyl)-6,8-dimethoxyiso coumarin (**341**)	*Xylaria* sp. hg1009	*Isodon sculponeatus*	-	[66]
Xylariaopyrones A–D (**342**–**345**)	*Xylariales* sp. (HM-1)	*Sophora tonkinensis*	Antibacterial activity	[115]
3*S*-Hydroxy-7melleine (**346**)	*Xylaria* sp. (No. 2508)	Mangrove	-	[116]
4′-*O*-Methyl-*β*-mannopyranoside (**347**)	*Xylaria feejeensis*	*Hintonia latiflora*	-	[117]
3*S*,4*R*-(+)-4-Hydroxymellein (**348**)	*Xylaria feejeensis*	*Hintonia latiflora*	*α*-glucosidase enzyme inhibitory activity	[117]
Xylarellein (**349**)	*Xylaria* sp. PSU-G12	*Garcinia hombroniana*	-	[118]
Xylariaindanone (**350**)	*Xylaria* sp. PSU-G12	*Garcinia hombroniana*	-	[118]
Xylapyrones A–F (**351**–**356**)	*Xylaria* sp. BM9	*Saccharum arundinaceum*	-	[119]
6-Heptanoyl-4-methoxy-2H-pyran-2-one (**357**)	*Xylaria* sp. GDG-102	*Sophora tonkinensis*	Antimicrobial activity	[120]
(+)-Phomalactone (**358**)	*Xylaria* sp. Grev. (Xylariaceae)	*Siparuna* sp.	Anti-plasmodial activity	[121]
6-(1-Propenyl)-3,4,5,6-tetrahydro-5-hydroxy-4Hpyran-2-one (**359**)	*Xylaria* sp. Grev. (Xylariaceae)	*Siparuna* sp.	-	[121]
Xylaolide A (**360**)	*Xylariaceae* sp. DPZ-SY43	Mangrove sediment	-	[122]
(*S*)-8-Hydroxy-6-methoxy-4,5-dimethyl-3-methyleneisochroman-1-one (**361**)	*Xylaria* sp. BL321	Mangrove-derived	-	[123]
(*R*)-7-hydroxy-3-((R)-1-hydroxyethyl)-5-methoxy-3,4-dimethylisobenzofuran-1(3H)-one (**362**)	*Xylaria* sp. BL321	Mangrove	-	[123]
Xylariaopyrone E-G (**363**–**365**)	*Xylaria* sp. (HM-1)	*Siparuna* sp.	Antimicrobial	[124]
Xylariaopyrone **H** (**366**)	*Xylaria* sp. (HM-1)	*Siparuna* sp.	-	[124]
Xylariaopyrone **I** (**367**)	*Xylaria* sp. (HM-1)	*Siparuna* sp.	Enzyme inhibitory activity	[124]
Lasobutone A (**368**)	*Xylaria* sp.	*Coptis chinensis*	-	[125]
Lasobutone B (**369**)	*Xylaria* sp.	*Coptis chinensis*	Anti-inflammatory activity	[125]
Coloratin A (**370**)	*Xylaria intracolorata*	Mangrove-derived	Antimicrobial activity	[126]
(3*R*)-6-Methoxy-5-methoxycarbonylmellein (**371**)	*Xylaria feejeensis*	*Geophila repensfungu*	Cytotoxicity	[93]
(3*S*,2′*R*,6′*R*)-Asperentin-8-*O*-methylether (**372**)	*Xylaria feejeensis*	*Geophila repensfungu*	Cytotoxicity	[93]
Xyolide (**373**)	*Xylaria feejeensis* (Berk.) Fr. (E6912B)	*Coptis chinensis*	Antifungal activity	[127]
Multiplolides A (**374**) and B (**375**)	*Xylaria multiplex* BCC 1111	Unknown	Antifungal activity	[128]
Xylariolides A–D (**376**–**379**)	*Xylaria* sp. NCY2	*Torreya jacki*	-	[106]
Furofurandione (**380**)	*Xylaria* sp. (BCC 21097)	*Licuala spinose*	-	[24]
3*R*,4*R*)-3,4-Dihydro-4,6-dihydroxy-3-methyl-1-oxo-1H-isochromene-5-carboxylic acid (**381**)	*Xylaria* sp.	Mushroom	Antifungal activity	[129]
(3*S*)-3,4-Dihydro-8-hydroxy-7-methoxy-3-methylisocoumarin (**382**)	*Xylaria* sp. SWUF09-62	*Ligustrum lucidum*	-	[108]
(3*S*)-3,4-Dihydro-5,7,8-trihydroxy-3-methylisocoumarin (**383**)	*Xylaria* sp. SWUF09-62	*Ligustrum lucidum*	Anti-inflammatory activity, cytotoxicity	[108]
4’-*O*-methyl-*β*-mannopyranoside (**384**)	*Xylaria feejeensis*	*Hintonia latiflora*		[117]
3*S*,4*R*-(+)-4-Hydroxymellein (**385**)	*Xylaria feejeensis*	*Hintonia latiflora*	Enzyme inhibitory activity	[117]
(3a*S*,6a*R*)-4,5-Dimethyl-3,3a,6,6a-tetrahydro-2H-cyclopenta [*β*]furan-2-one (**386**)	*Xylaria curta* 92092022	Unknown	Antibacterial activity	[111]
Macrolide (**387**)	*Xylaria feejeensis*	*Piper aduncum*	Anti-osteoporosis activity	[130]
(−)-Annularin C (**388**)	*Xylaria feejeensis*	*Piper aduncum*	-	[130]
Xylarone (**389**)	*Xylaria hypoxylon* A27-94	Unknown	Anti-proliferative	[131]
8,9-Dehydroxylarone (**390**)	*Xylaria hypoxylon* A27-94	Unknown	-	[131]
Rubiginosic acid (**391**)	*Xylariaceus ascomycete*	*Corylus avellana*	-	[102]
Xylarianins A, C and D (**392**–**394**)	*Xylaria* sp. SYPF 8246	*Panax notoginseng*	-	[85]
6-Methoxycarbonyl-2′-methyl-3,5,4′,6′-tetramethoxy-diphenyl ether (**395**)	*Xylaria* sp. SYPF 8246	*Panax notoginseng*	-	[85]
2-Chlor-6-methoxycarbonyl-2′-rnethyl-3,5,4′,6′- tetramethoxy-diphenyl ether (**396**)	*Xylaria* sp. SYPF 8246	*Panax notoginseng*	-	[85]
2-Chlor-4′-hydroxy-6-methoxy carbonyl-2′-methyl-3,5,6′-trimethoxy-diphenyl ether (**397**)	*Xylaria* sp. SYPF 8246	*Panax notoginseng*	-	[85]
Akoenic acid (**400**)	*Xylaria* cubensis	Leaves of *L. akoensis* Hayata (Lauraceae)	-	[11]
Phenethylester (**401**)	*Xylaria nigripes* (Kl.) Sacc	*Garcinia hombroniana*	Cytotoxicity	[73]
3,7-Dimethyl-9-(-2,2,5,5-tetramethyl-1,3-dioxolan-4-yl) nona-1,6-dien-3-ol (**402**)	*Xylaria* sp.	*Taxus mairei*	Antibacterial activity	[132]
Rubiginosic acid (**403**)	*Xylaria ascomycete*	*Fraxinus excelsior*	-	[100]
Xylarosides A (**404**) and B (**405**)	*Xylaria* sp. PSU-D14	*Garcinia dulcis*	-	[133]
2-Methyl-2-(4-hydroxymethylphenyl) oxacyclopentane (**406**)	*Xylaria polymorpha* (Pers.: Fr.)	Dead branch	Antifungal activity	[13]
Penixylarins A and D (**407** and **410**)	*Penicillium crustosum* PRB-2 and *Xylaria* sp. HDN13-249	Antarctic deep sea	-	[134]
Penixylarins B and C (**408** and **409**)	*Penicillium crustosum* PRB-2 and *Xylaria* sp. HDN13-249	Antarctic deep sea	Antibacterial activity	[134]
Phenylacetic acid derivative (**411**)	*Xylaria* sp. FM1005	*Sinularia densa*	-	[69]
Naphthalenedicarboxylic acid (**412**)	*Xylaria* sp. FM1005	*Sinularia densa*	-	[69]
Xylatriol (**413**)	*Xylaria* sp.	-	-	[89]
Acumifurans A–C (**414**–**416**)	*X. acuminatilongissima* YMJ623	*Odontotermes formosanus*	-	[135]
(2*E*,4*E*,6*S*)-6-Hydroxydeca-2,4-dienoic acid (**417**)	*Xylaria* sp. C-2	*Gorgonian*	-	[136]
(24*R*)-22,23-Dihydroxy-ergosta-4,6,8(14)-trien-3-one 23-*β*-*D*-glucopyranoside (**418**)	*Xylaria* sp.	Mangrove	Cytotoxicity	[137]
Xylarester (**419**)	*Xylaria* sp.	Mangrove	-	[137]
Coloratin B (**420**)	*Xylaria intracolorata*	Mushroom	-	[119]
Xylaril acids A–C (**421**–**423**)	*Xylaria longipes*	*Termite nest*	Neuroprotective activity	[138]
Xylaril acids D and E (**424** and **425**)	*Xylaria longipes*	*Termite nest*	Neuroprotective activity	[138]
Wheldone (**426**)	*Aspergillus fischeri* (NRRL 181) and *Xylaria flabelliformis* (G536)	*Termite nest*	Cytotoxicity	[139]
Fimbriether A, C, D, and F (**427**, **429**, **430,** and **432**)	*Xylaria fimbriata* Lloyd (YMJ491)	*Termite nest*	-	[140]
Fimbriethers B and E (**428** and **431**)	*Xylaria fimbriata* Lloyd (YMJ491)	*Termite nest*	Anti-inflammatory activity	[140]
Xylarioic acid B (**434**)	*Xylaria* sp. NCY2	*Torreya jacki*	-	[106]
Xylariate C (**435**)	*Xylaria* sp. NCY2	*Torreya jacki*	-	[106]
Xylopimarane (**436**)	*Xylaria* sp. (BCC 4297)	Mushroom	Cytotoxicity	[141]
Xylarinic acids A (**437**) and B (**438**)	*Xylaria polymorpha* (Pers.) Grev	Fruit body	Antifungal activity	[142]
Xylacinic acids A (**439**) and B (**440**)	*Xylaria cubensis* PSU-MA34	*Termite nest*	-	[143]
Allantoside (**441**)	*Xylaria allantoidea* SWUF76.	Unknown	-	[68]
Ergosta-4,6,8(14),22-tetraen-3-one (**442**)	*Xylaria* sp.	Unknown	Anti-inflammatory activity	[144]
Xylarosides A (**443**) and B (**444**)	*Xylaria* sp. PSU-D14	Leaves of *Garcinia dulcis*	-	[133]
Aminobenzoate (**445**)	*Xylaria* sp. BCC 9653	Wood-decayed	-	[145]

## Data Availability

Not applicable.

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
