# Peer review of "Structures and Biological Activities of Secondary Metabolites from Xylaria spp."

_jof, 2024, doi:10.3390/jof10030190_

Round 1

Reviewer 1 Report

The authors of this review report that «The secondary metabolites of the fungus genus Xylaria have not been summarized in systematically or detail».

However, there are two review articles  2022 (Deshmukh, S.K., Sridhar, K.R., Saxena, S., Gupta, M.K. (2022). Recent Advances in the Discovery of Bioactive Metabolites from Xylaria Hill ex Schrank. In: Arya, A., Rusevska, K. (eds) Biology, Cultivation and Applications of Mushrooms. Springer, Singapore. https://doi.org/10.1007/978-981-16-6257-7_3)  and 2017 (Macías-Rubalcava ML, Sánchez-Fernández RE. Secondary metabolites of endophytic Xylaria species with potential applications in medicine and agriculture. World J Microbiol Biotechnol. 2017 Jan;33(1):15. doi: 10.1007/s11274-016-2174-5. Epub 2016 Nov 28. PMID: 27896581.) which presents material on bioactive metabolites of fungi of the genus Xylaria. Years of scientific literature coverage from 1994 to 2022.

In this regard, the authors of this review article need to familiarize themselves with the material published on this topic and edit their review taking into account the available information.

In my opinion, repetitions should be excluded or described briefly with reference to well-known reviews. This review should focus on information that is not provided in published review articles.

Information should also be added on possible areas of application and purpose of bioactive metabolites of fungi of the genus Xylaria.

Page 1 Line 29, 33, 35  Xylarial replace with Xylaria

Author Response

The authors of this review report that «The secondary metabolites of the fungus genus Xylaria have not been summarized in systematically or detail». However, there are two review articles  2022 (Deshmukh, S.K., Sridhar, K.R., Saxena, S., Gupta, M.K. (2022). Recent Advances in the Discovery of Bioactive Metabolites from Xylaria Hill ex Schrank. In: Arya, A., Rusevska, K. (eds) Biology, Cultivation and Applications of Mushrooms. Springer, Singapore. https://doi.org/10.1007/978-981-16-6257-7_3)  and 2017 (Macías-Rubalcava ML, Sánchez-Fernández RE. Secondary metabolites of endophytic Xylaria species with potential applications in medicine and agriculture. World J Microbiol Biotechnol. 2017 Jan;33(1):15. doi: 10.1007/s11274-016-2174-5. Epub 2016 Nov 28. PMID: 27896581.) which presents material on bioactive metabolites of fungi of the genus Xylaria. Years of scientific literature coverage from 1994 to 2022.

Q1: In this regard, the authors of this review article need to familiarize themselves with the material published on this topic and edit their review taking into account the available information.

In my opinion, repetitions should be excluded or described briefly with reference to well-known reviews. This review should focus on information that is not provided in published review articles.

A: Thank you for your comments. We have revised our manuscript according to your advice.  The published two review articles have reported the bioactive compounds from Xylaria species articles, and the cited reference are as to 2020. The published reviews focus on the bioactive metabolites (new and known compounds), as of 2020, 245 bioactive compounds (118 new compounds) has been reported in the reference.

While we focus on the secondary metabolites with new structures (including the structure types) and their bioactivities. So we have deleted the known bioacitve compounds, described briefly with reference to well-known reviews., and added the reference of the new compounds as to January 2024. A total number of 445 new compounds, including terpenoids, nitrogen-containing compounds, polyketides, lactones, and other classes, are presented in our review. Q2: Information should also be added on possible areas of application and purpose of bioactive metabolites of fungi of the genus Xylaria.

A: Thank you for your comments. We add the possible areas of application in the content of Introduction, and add the purpose of bioactive metabolites of fungi of the genus Xylaria in the content of Conclusion.

Q3: Page 1 Line 29, 33, 35  Xylarial replace with Xylaria

A: Thank you for your comments. We have revised the Xylarial as Xylaria.

Reviewer 2 Report

Although the presented information is interesting, as his manuscript presents relevant information and comprehensive revision on the bioactive compounds reported for Xylaria spp, several modifications must be done to be considered for publication in this prestigious Journal.

Criteria used to build this review must be described.

Review articles should clearly describe and discuss the main findings supported by other studies with a strict scientific focus on why these results are relevant to other similar studies. The discussion and critical contribution of this manuscript does not meet the scientific requirements to be published in this prestigious Journal. Thus, an explanation about how the bioactive compounds reported in Xylaria spp. exhibit the bioactivities reviewed should be included into each subsection (as appropriate). Some examples are described below; however, whole manuscript must be revised in the same way.

Lines 56 – 57: How does this happen? Explain

Lines 60 – 62: How does this inhibition take place?

Lines 73: 75: Describes this antimicrobial mechanism

Lines 85 – 86: How does this happen? Explain

Lines 100 – 101: How does this compound inhibit such enzyme?

Lines 106 – 107: How does this immunosuppressive activity occur?

Lines 111 – 118: How does this antimicrobial activity take place?

And so on

Conclusion must be added

Author Response

Although the presented information is interesting, as his manuscript presents relevant information and comprehensive revision on the bioactive compounds reported for Xylaria spp, several modifications must be done to be considered for publication in this prestigious Journal. Criteria used to build this review must be described.

Q1: Review articles should clearly describe and discuss the main findings supported by other studies with a strict scientific focus on why these results are relevant to other similar studies. The discussion and critical contribution of this manuscript does not meet the scientific requirements to be published in this prestigious Journal. Thus, an explanation about how the bioactive compounds reported in Xylaria spp. exhibit the bioactivities reviewed should be included into each subsection (as appropriate). Some examples are described below; however, whole manuscript must be revised in the same way.

Lines 56 – 57: How does this happen? Explain

Lines 60 – 62: How does this inhibition take place?

Lines 73: 75: Describes this antimicrobial mechanism

Lines 85 – 86: How does this happen? Explain

Lines 100 – 101: How does this compound inhibit such enzyme?

Lines 106 – 107: How does this immunosuppressive activity occur?

Lines 111 – 118: How does this antimicrobial activity take place?

And so on

A: Thank you for your comments. We have revised the above questions in our revised manuscript.

Q2: Conclusion must be added.

A: Thank you for your comments. We have added the content of Conclusion.

Reviewer 3 Report

This paper describes a review of compounds and bioactivities described for Xylaria species comprising from 1996 to 2023.  This important revision covers recent years without reviews for the genus.

The abstract has some mistakes in the language and it should be revised.

The name of genus is Xylaria instead Xylarial. This error should be corrected in many citations of the text.

Structures 223 & 224 are not polyketides. These structures are eremophilane sesquiterpenes. These compounds should be reorganized in the text and figures.  

Table 1. The habitats are related to the fungi species. They are not related to the metabolite as the authors named the table. The title of the table should be revised.    

Author Response

This paper describes a review of compounds and bioactivities described for Xylaria species comprising from 1996 to 2023.  This important revision covers recent years without reviews for the genus.

Q1: The abstract has some mistakes in the language and it should be revised.

A: Thank you for your comments. We have revised mistakes in the language in Abstract in our revised manuscript.

Q2: The name of genus is Xylaria instead Xylarial. This error should be corrected in many citations of the text.

A: Thank you for your comments. We have changed the Xylaria as Xylarial in our revised manuscript.

Q3: Structures 223 & 224 are not polyketides. These structures are eremophilane sesquiterpenes. These compounds should be reorganized in the text and figures.  

A: Thank you for your comments. Our manuscript focus on the new compounds isolated from the  Xylaria  fungi, so we have deleted the Structures 223 & 224 in our revised manuscript.

Q4: Table 1. The habitats are related to the fungi species. They are not related to the metabolite as the authors named the table. The title of the table should be revised.    

A: Thank you for your comments. We have revised Table 1 in our revised manuscript.

Round 2

Reviewer 2 Report

No comments

No comments